# Economic Emission Dispatch Considering Renewable Energy Resources—A Multi-Objective Cross Entropy Optimization Approach

**Qun Niu [1,\*], Ming You [1], Zhile Yang [2] and Yang Zhang [1]**

[1]  School of Mechanical Engineering and Automation, Shanghai University, Shanghai 200444, China; youming1227@shu.edu.cn (M.Y.); zy2016@shu.edu.cn (Y.Z.)

[2]  Shenzhen Institute of Advanced Technology, Chinese Academy of Sciences, Shenzhen 518055, China; zyang07@qub.ac.uk

\*  Correspondence: nq@shu.edu.cn

**Abstract:** The conventional electrical power system economic dispatch (ED) often only pursues immediate economic benefits but neglects the harmful environment impacts of gas emissions from thermal power plants. To address this shortfall, economic emission dispatch (EED) has drawn a lot of attention in recent years. With the increasing penetration of renewable generation, the intermittence and uncertainty of renewable energy such as solar power and wind power increase the difficulties of power system scheduling. To enhance the dispatch performance with significant penetration of renewable energy, a modified multi-objective cross entropy algorithm (MMOCE) is proposed in this paper. To solve multi-objective optimization problems, a crowding–distance calculation technique and a novel external archive mechanism are introduced into the conventional cross entropy method. Additionally, the population updating process is simplified by introducing a self-adaptive parameter operator that substitutes the smoothing parameters, while the solution diversity and the adaptability in large scale systems are improved by introducing the crossover operator. Finally, a two-stage evolutionary mechanism further enhances the diversity and the rate of convergence. To verify the efficacy of the proposed MMOCE, eight benchmark functions and three different test systems considering different mixes of renewable energy sources are employed. The dispatch results by the proposed MMOCE are compared with other multi-objective cross entropy algorithms and published heuristic methods, confirming the superiority of the proposed MMOCE over other methods in all test systems.

**Keywords:** economic emission dispatch; renewable energy sources; multi-objective cross entropy algorithm; crossover operator

## 1. Introduction

The economic dispatch (ED) is a fundamental issue in electrical power system scheduling that aims to maximize the economic profits by the optimal allocation of the output of each generator unit [1]. It is shown that more than 10% of the total energy consumption can be saved by means of economic dispatch [2]. However, in the last decades, environmental issues have drawn substantial attention worldwide, and much effort has been made to mitigate the negative effects of climate change and environment pollution [3]. As such, managing pernicious gases emissions has become an important consideration when it comes to the ED problem. The economic emission dispatch (EED) retains the original characteristics while incorporating the emission factors, resulting in a multi-objective optimization problem, which simultaneously minimizes the generating cost and the emission level to the lowest possible values.

In China, about 70% of electricity supply comes from coal-fired power stations; addressing climate change while meeting future energy needs will inevitably impose greater

pressures on traditional energy sources [4]. Today, high penetration of renewable energy sources is expected for electric power systems, and many attempts to integrate renewable energy resources into the EED problems are reported [5]. As of 2015, more than 9% of electricity generation comes from renewable resources [6]. Renewable energy refers to the resources that can be used repeatedly to produce energy, such as solar energy, wind energy, biomass energy, geothermal energy, etc., [7]. It is also often referred to as alternative energy. Utilizing abundant renewable energy resources reduces the emission of harmful gases to a great extent. The majority of studies revealed that integrating renewable energy resources such as wind energy conversion systems (WECS) and PV solar systems into the power system is a promising technology and practically feasible. Nevertheless, the existing grid faces unpredictable and intermittent power supply from the resources, particularly from wind and photovoltaic solar. The electric power production from renewable energy resources such as wind speed and solar radiation can be very indeterminate, variable with time, and undispatchable with limited control, imposing major challenges on the scheduling of power systems [8]. Their intermittent power output imposes various difficulties on the scheduling, operation, and control of the power system networks. Therefore, handling renewable energy resources requires sophisticated planning and operation scheduling as well as state-of-the-art technologies.

As a research hotspot in the field of power system optimization, some progress has been reported on solving the EED problems incorporating renewable energy resources in recent years. In [9], the multi-objective economic emission dispatch problem is considered, which combines heat and wind power generation in a large micro-grid (MG), and IEEE 30-bus and 69-bus systems are used to represent a large MG. In [10], a solar–wind–thermal ELD problem integrating a pumped-storage hydraulic unit is considered. In [11], there is an analysis of the economic dispatch and unit commitment problem where five scenarios are considered to predict the power system operations by 2030. In [12], the EED model of an IEEE 30-bus system incorporating solar photovoltaic, wind and battery storage is built and multiple objectives are considered. The authors of [13] present an EED model that integrates thermal, natural gas, and renewable energy systems, considering both emission levels and generation cost. These reports describe the EED model considering renewable energy resources in detail, and various optimization strategies are utilized. Since there exist a number of EED problem types with renewable energy, most studies only consider one type of EED model, and the proposed methods may be difficult to apply to other kind of models. It is therefore meaningful to consider multiple types of EED models with different renewable energy sources. Under this consideration, this paper will investigate three different types of EED problems integrated renewable energy resources.

For the EED problem, a number of algorithms have been attempted so far. Generally speaking, these algorithms can be mainly grouped into two categories [14,15]. In the first category, all objectives are combined linearly so that the EED problem can be regarded as a single objective optimization problem [16–18]. Meanwhile, given the nonlinear and nonconvex nature of the dispatch problem, the classical deterministic techniques such as weighted sum method, $\epsilon-$constraints, and linear programming may fail to solve the problem due to the sheer computational complexity of the problem, inclination to fall into local optima, and discontinuities and nonsmoothness in the process of handling functions. Hence, nature-inspired metaheuristic algorithms have attracted substantial interests. The drawbacks of this approach, however, are multi-faceted: First, only one solution is generated in a single run, so it is time-consuming. Second, the exact weight parameters in the objective function are rather difficult to optimize.

The second category of algorithms is to deal with the two objectives in the EED problem simultaneously. The Pareto front can be easily obtained with a single run so that the optimal solution can be chosen according to different principles. Many Pareto-based multi-objective algorithms have been adopted to solve the EED problem successfully, such as NSGAII, MOPSO, and SPEA. Qiao [19] employed a self-adaptive multi-objective differential evolution algorithm on a novel dynamic economic emission dispatching framework



integrating both electric vehicles and wind farms and demonstrated that the proposed method can improve the results efficiently in different test systems based on 10-unit generators. Habibi [20] applied a multi-objective particle swarm optimization (MOPSO) algorithm by optimizing the objective function of the storage and additional cost to solve the EED problem incorporating stochastic wind in multi-area power systems. Jiang [21] used gravitational particle swarm optimization algorithm (GPSOA) to solve the EED problem considering wind power availability for the wind-thermal power system. Li [22] established a dynamic economic emission scheduling model incorporating wind power, solar power, and hydropower under tradable green certificate and proposed a new multi-objective moth-flame optimization (MOMFO) to resolve the model. Liao [23] presented a multi-objective optimization by learning automata (MOLA) and implemented it on the EED model such that the objective functions are formulated as a combination of cost, emission and voltage stability. Ghasemi [24] developed a new Honey Bee Mating Optimization (HBMO) with blended online learning mechanism for the EED model with uncertain wind power. However, these methods may still fail to obtain better results despite of their complexity or they have several parameters to pre-determine in the process of optimization that may greatly affect the optimization results. It is therefore worthwhile to develop a simple yet efficient algorithm to solve multiple objective optimization problems.

The cross entropy (CE) method proposed by Robinstein [25,26] is an innovative meta-heuristic algorithm based on rare events resampling and the Kullback–Leibler distance minimization. The main idea of the CE method is to select elite samples from the best performing populations and generate new populations according to the distribution of elite samples. The superiority of CE lies in its diversified structure and its simple process of updating populations. Hence, the CE method has been successfully applied to a range of engineering problems [27–29] including the ED problem [30]. In [31], the CE method was firstly extended to multi-objective optimization and achieved good performance. In recent years, the multi-objective cross entropy method has been further improved and applied to a range of problems, such as finance [32], computer network [33], and engineering design [34]. For example, in [35], a cascaded algorithm combining the cross-entropy (CE) and Tabu search (TS) was successfully applied to building an effective boiler efficiency prediction model and improved the economic benefit and reliability of generating units. Dorini et al. [36] proposed a novel algorithm based on the noisy cross-entropy sensor locator (nCESL) to detect accidental and/or intentional contamination in water distribution systems. Sun et al. [37] improved the CE method and used it to solve the multi-objective energy routes problem in a WPTN system. Perelman et al. [38] extended the conventional CE method to multi-objective combinatorial optimization of water distribution systems design and demonstrated its better robustness comparing with NSGAII. Bekker et al. [39] adapted the CE method to multi-objective optimization and applied the proposed algorithm to a dynamic, stochastic problem and demonstrated its effectiveness and efficiency. However, it is worth mentioning that only limited papers reported to have applied the multi-objective CE methods to solving the EED problems so far. For instance, in [40], a cross entropy optimization based on decomposition was proposed to solve a multi-objective optimization problem for a model of a wind/hydro/thermal/photovoltaic power system with 10 generators. However, the performance of most CE methods tends to be less optimistic in solving relatively large-scale EED problems. In addition, they have a number of smoothing parameters, which may cause premature convergence and limit the algorithm performance. Therefore, it is necessary to improve the design of the multi-objective cross entropy algorithm with far fewer smoothing parameters and improve their applicability in solving complex EED problems.

As mentioned earlier, the conventional CE method has the advantage of a versatile structure and simple updating procedure. Nevertheless, the drawbacks of the CE method are also evident: they easily fall into local optimum and need to predetermine several constant parameters such as the smoothing parameters (different values of constant parameters may result in different optimization performance). While in the field of multi-

objective optimization, the performances of most existing multi-objective CE methods are still yet to be satisfactory when dealing with large scale problems. In this paper, a modified multi-objective cross entropy algorithm (MMOCE) is presented to solve EED problems considering the presence of renewable energy sources. First, a self-adaptive operator substitutes the smoothing parameters in the updating formula for the mean value and the standard deviation. This new updating process is proposed to reduce the number of pre-defined constant parameters, which makes the algorithm simpler and more stable. Second, the crossover operator is employed in present population and the external archive to improve the global search mechanism and the scalability to large-scale test systems. Third, different parameter evolutionary mechanisms as stated in [41], which are implemented to calculate the mean value and the standard deviation of the present population according to population generation, are introduced into MMOCE to enhance the diversity of solutions and to further speed up the convergence. With these improvements, the algorithm efficiency is greatly enhanced. Eight benchmark functions and two performance indicators are used to verify the feasibility and superiority of the proposed MMOCE by comparing with other state-of-the-art algorithms such as NSGAII, MOPSO, MOEA/D, and PESA2. Additionally, in this paper, three different types of EED problems with different scales and combinations of renewable energy resources are introduced to testify the superiority of the proposed method. The first problem is a combined heat, emission, and economic dispatch (CHEED) problem integrating wind and solar power generations. The second problem is an IEEE 30-bus and six-generator system that incorporates stochastic wind and solar power. The third problem is a 40-unit combined emission economic dispatch problem with wind penetration. The results obtained from the above three examples confirm that the proposed method outperforms other competing algorithms.

## 2. Mathematical Model

This paper mainly focuses on three different environmental economic dispatch models considering renewable sources, including the combined heat, emission, and economic dispatch (CHEED), and applies the proposed method to a modified IEEE 30-bus and six-generator system with renewable energy and combined emission economic dispatch problems with wind penetration. The overall dispatching framework of the first model are shown in Figure 1.

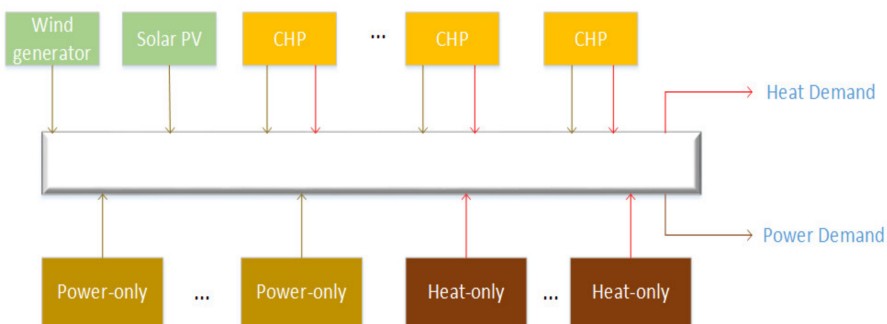

**Figure 1.** The CHEED system.

### 2.1. Problem Formulation of CHEED

The system considered in this paper contains conventional generators, cogeneration units, and heat-only units. The feasible operating region (FOR) of the cogeneration units is illustrated in Figure 2 [42], which is enclosed by the boundary curve ABCDEF. For the CHEED problem, the power is derived from the thermal units and cogeneration units while the heat is derived from cogeneration units and heat-only units. Under the condition of ensuring the power balance of the system and satisfying the constraints, the output of each unit is reasonably allocated to achieve the goal of minimizing the cost of heat and power

production while minimizing the emission level. Thus, CHEED can be mathematically elaborated as follows.

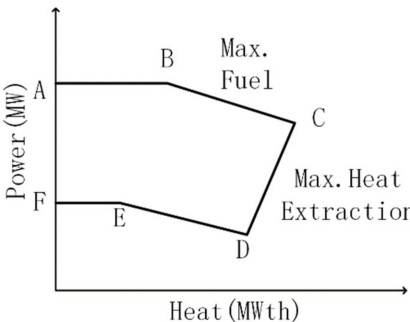

**Figure 2.** Feasible operation region for a cogeneration unit.

### 2.1.1. Objective Functions

1.  Total cost is defined as:

$$
\begin{aligned}
C_{\text{total}} &= \sum_{i=1}^{N_n} C_i(P_i) + \sum_{j=1}^{N_c} C_j(P_j, H_j) + \sum_{k=1}^{N_h} C_k(H_k) \\
&= \sum_{i=1}^{N_n} \left( a_i + b_i P_i + d_i P_i^2 \right) \\
&+ \sum_{j=1}^{N_c} \left( m_j + n_j P_j + l_j P_j^2 + x_j P_j H_j + y_j H_j + z_j H_j^2 \right) \\
&+ \sum_{k=1}^{N_h} \left( \alpha_k + \beta_k H_k + \gamma_k H_k^2 \right)
\end{aligned}
\tag{1}
$$

2.  Emission is defined as:

$$
\begin{aligned}
E_{\text{total}} &= \sum_{i=1}^{N_p} E_{pi}(P_i) + \sum_{j=1}^{N_c} E_{cj}(P_j, H_j) + \sum_{k=1}^{N_h} E_{hk}(H_k) \\
&= \sum_{i=1}^{N_n} \left( \delta_i + \varepsilon_i P_i + \xi_i P_i^2 \right) + \sum_{j=1}^{N_c} \mu_j P_j + \sum_{k=1}^{N_h} \sigma_k H_k
\end{aligned}
\tag{2}
$$

### 2.1.2. Constraints

In this paper, the renewable energy is incorporated into the CHEED problem and plays the role of negative loads in order to decrease the demand load. Simultaneously, the electricity and heat production and capacity of each unit are subject to their respective constraints.

$$
\sum_{i=1}^{N_p} P_i + \sum_{j=1}^{N_c} P_j + P_{solar} + P_{wind} = P_d + P_L
\tag{3}
$$

$$
\sum_{j=1}^{N_c} H_j + \sum_{k=1}^{N_k} H_k = H_D
\tag{4}
$$

$$
P_i^{min} \leq P_i \leq P_i^{max}, \ i \in 1, 2, \dots, N_p
\tag{5}
$$

$$
H_k^{min} \leq H_k \leq H_k^{max}, \ k \in 1, 2, \dots, N_k
\tag{6}
$$

$$
P_j^{min}(H_j) \leq P_j \leq P_j^{max}(H_j), \ j \in 1, 2, \dots, N_c
\tag{7}
$$

$$
H_j^{min}(P_j) \leq H_j \leq H_j^{max}(P_j), \ j \in 1, 2, \dots, N_c
\tag{8}
$$

The power transmission loss can be calculated by B-matrix coefficients as follows [43]:

$$P_L = \sum_{i=1}^{N_p+N_c} \sum_{j=1}^{N_p+N_c} P_i B_{ij} P_j + \sum_{i=1}^{N_p+N_c} B_{oi} P_i + B_{oo} \tag{9}$$

### 2.2. Mathematical Models of the Modified IEEE 30-Bus and Six-Generator System with Wind and Solar Energy

### 2.2.1. Cost of Conventional Thermal Units

The overall dispatching framework of this model are shown in Figure 3. The cost of conventional thermal units primarily refers to fossil fuel cost. Meanwhile, in order to make the cost function more accurate and realistic, the valve effect, which is formulated as a sinusoidal function and added to the basic cost function, is also taken into consideration. Then, the ultimate cost function of convention thermal units can be expressed as follows in Equation (10).

$$C_{To}(P_{TU_j}) = \sum_{j=1}^{N_{TU}} \left( \alpha_j + \beta_j P_{TU_j} + \gamma_j^2 P_{TU_j}^2 + \left| d_j \times \sin\left(e_j \times \left(P_{TU_j}^{min} - P_{TU_j}\right)\right)\right| \right) \tag{10}$$

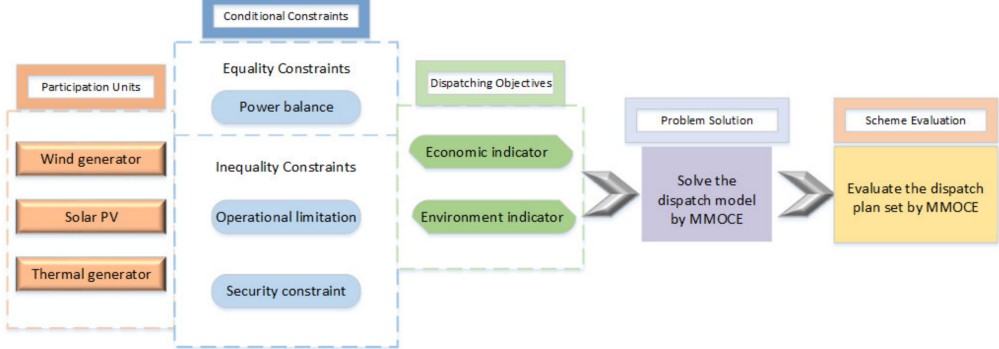

**Figure 3.** The overall dispatching framework of the IEEE 30-bus system.

### 2.2.2. Cost of Wind Energy

As one of the most common renewable energy resources, wind power is characterized by its intermittency and uncertainty. Provided that wind farms are unable to provide enough available scheduled power, an independent system operator (ISO) will take the responsibility of handling the deficit in order to maintain spinning reserves. This case is deemed as overestimation of renewable energy and the power generation cost will have to increase in order to maintain the spinning reserves. In contrast, if these farms provide excessive power, the ISO should accept the penalty. This case is called underestimation. In general, the total cost of wind power includes direct cost, reserve cost, and penalty cost [44].

The direct cost of wind power can be expressed as follows [45].

$$C_{\text{wind, } i}(P_{sw,i}) = h_i P_{sw,i} \tag{11}$$

In addition, the situations of overestimation and underestimation may always exist and thus increase the overall costs. In summary, overestimation will lead to the reserve cost and underestimation will lead to the penalty cost. They can be described, respectively, as follows [45].

$$\begin{aligned} C_{RC,i}(P_{sw,i} - P_{wa,i}) &= K_{RC,i}(P_{sw,i} - P_{wa,i}) \\ &= K_{RC,i} \int_0^{P_{sw,i}} (P_{sw,i} - P_{w,i}) f_{wp}(P_{w,i}) dP_{w,i} \end{aligned} \tag{12}$$

$$C_{PC,i}(P_{wa,i} - P_{sw,i}) = K_{PC,i}(P_{wa,i} - P_{sw,i})$$
$$= K_{PC,i} \int_{P_{sw,i}}^{P_{rw,i}} (P_{w,i} - P_{sw,i}) f_{wp}(P_{w,i}) dP_{w,i} \tag{13}$$

In this paper, the Weibull probability density function (PDF) [46,47] is used to represent the wind speed distribution and Figure 4a illustrates the Weibull PDF of the wind speed data by employing 8000 Monte-Carlo scenarios. Thus, the probability of wind speed can be described as below, where $c$ is the scale factor and $k$ is a shape factor.

$$f_v(v) = \left(\frac{k}{c}\right)\left(\frac{v}{c}\right)^{k-1} e^{-\left(\frac{v}{c}\right)^k}, v \geq 0 \tag{14}$$

The mean of the Weibull distribution is formulated as:

$$\begin{cases} M_{wbl} = c \times \Gamma\left(1 + k^{-1}\right) \\ \Gamma(x) = \int_0^\infty e^{-t} t^{x-1} dt \end{cases} \tag{15}$$

The power output of a wind turbine is approximately defined as follows:

$$P_w(v) = \begin{cases} 0, v < V_{in} \text{ and } v > V_{out} \\ P_{rw}\left(\frac{v - V_{in}}{V_r - V_{in}}\right), V_{in} \leq v \leq V_r \\ P_{rw}, V_r < v \leq V_{out} \end{cases} \tag{16}$$

Next, wind power probabilities can be piecewise calculated by [48]:

$$f_{wp}(P_w)\{P_w = 0\} = 1 - \exp\left[-\left(\frac{V_{in}}{c}\right)^k\right] + \exp\left[-\left(\frac{V_{out}}{c}\right)^k\right] \tag{17}$$

$$f_{wp}(P_w)\{P_w = P_{rw}\} = \exp\left[-\left(\frac{V_r}{c}\right)^k\right] - \exp\left[-\left(\frac{V_{out}}{c}\right)^k\right] \tag{18}$$

$$f_w(p_w) = \frac{k(V_r - V_{in})}{c^k \times p_{rw}}\left[V_{in} + \frac{p_w}{p_{rw}}(V_r - V_{in})\right]^{k-1} \times \exp\left[-\left(\frac{V_{in} + \frac{p_w}{p_{rw}}(V_r - V_{in})}{c}\right)^k\right] \tag{19}$$

### 2.2.3. Cost of Solar Photovoltaic Power

As the most popular renewable source, similar to wind power, the total cost of solar power also contains direct cost, reserve cost, and penalty cost. Here, the direct cost involved with solar power can be computed as:

$$C_{\text{solar},j}(P_{ss,j}) = g_j P_{ss,j} \tag{20}$$

Reserve cost for the solar power plant is defined as:

$$C_{SR,j}(P_{ss,j} - P_{sa,j}) = K_{SR,j}(P_{ss,j} - P_{sa,j})$$
$$= K_{SR,j} \times f_s(P_{sa,j} < P_{ss,j}) \times [P_{ss,j} - E(P_{sa,j} < P_{ss,j})] \tag{21}$$

where $f_s(P_{sa,j} < P_{ss,j})$ denotes the probability of event of solar power shortage associated with the scheduled power $P_{ss,j}$. $E(P_{sa,j} < P_{ss,j})$ denotes the expectation of solar power below $P_{ss,j}$. Penalty cost for underestimation is defined as [45]:

$$C_{PS,j}(P_{sa,j} - P_{ss,j}) = K_{PS,j}(P_{sa,j} - P_{ss,j})$$
$$= K_{PS,j} \times f_s(P_{sa,j} > P_{ss,j}) \times [E(P_{sa,j} > P_{ss,j}) - P_{ss,j}] \tag{22}$$

Similarly, $f_s(P_{sa,j} > P_{ss,j})$ denotes the probability of event of solar power shortage associated with the scheduled power $P_{ss,j}$. $E(P_{sa,j} > P_{ss,j})$ denotes the expectation of the solar power above $P_{ss,j}$.

In this paper, the Longnormal PDF is illustrated in Figure 4b and adopted to describe solar irradiance ($G_{sr}$). According to the Longnormal PDF with mean $\mu$ and standard deviation $\varsigma$, the probability of solar irradiance can be calculated as follows [49]:

$$f_{G_{ss}}(G_{sr}) = \frac{1}{G_{sr}\delta\sqrt{2\pi}} \exp\left\{\frac{-(\ln x - \mu)^2}{2\delta^2}\right\}, G_{sr} > 0 \tag{23}$$

The mean of Lognormal distribution can be expressed as:

$$M_{\lg n} = \exp\left(\mu + \frac{\sigma^2}{2}\right) \tag{24}$$

Based on the analysis from [45], the overestimation cost of Equation (21) is calculated as:

$$
\begin{aligned}
C_{SR}(P_{ss} - P_{sa}) &= K_{SR}(P_{ss} - P_{sa}) \\
&= K_{SR}\sum_{n=1}^{N^-}[P_{ss} - P_{sn-}] \times f_{sn-}
\end{aligned}
\tag{25}
$$

where $P_{sn-}$ represents the relevant power less than the scheduled power. $f_{sn-}$ denotes the relative frequency of occurrence of $P_{sn-}$. $N^-$ stands for the number of pairs ($P_{sn-}$, $f_{sn-}$) generated for the PDF. According to [45], the underestimation cost of Equation (22) can be computed as:

$$C_{PS}(P_{sa} - P_{ss}) = K_{PS}\sum_{n=1}^{N^*}[P_{sn+} - P_{ss}] \times f_{sn+} \tag{26}$$

where $P_{sn+}$ represents the relevant power more than the scheduled power. $f_{sn+}$ denotes the relative frequency of occurrence of $P_{sn+}$. $N^*$ is the number of pairs ($P_{sn+}$, $f_{sn+}$) generated for the PDF.

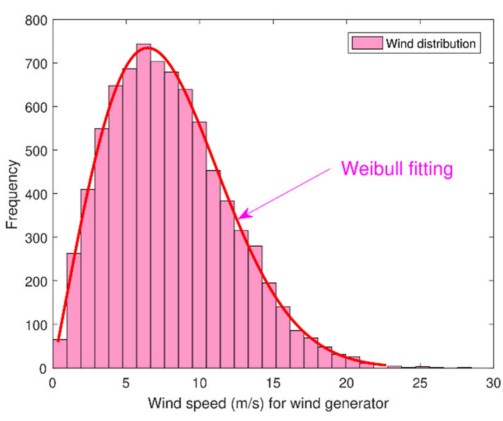
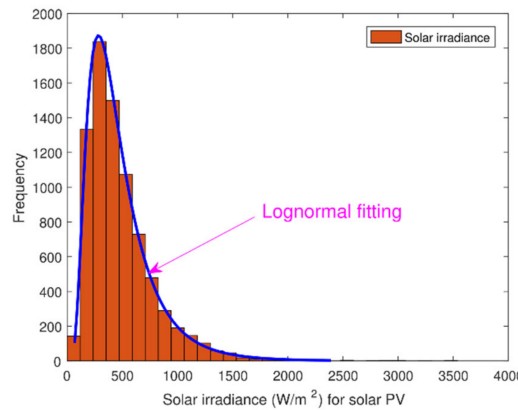

(**a**)Weibull fitting of wind distribution  (**b**)Lognormal fitting of solar irradiance

**Figure 4.** Wind speed distribution and solar irradiance distribution.

### 2.2.4. Emission Function

It is generally understood that harmful gases (mainly $SO_x$ and $NO_x$) will be emitted into the atmosphere when electric power is generated from conventional fossil fuels. The functional relationship between pollutants emissions and the produced power from conventional thermal generators is defined as:

$$E(P_{TU}) = \sum_{i=1}^{N_{TU}}\left[\left(p_i + q_iP_{TU_i} + r_iP_{TU_i}^2\right) \times 0.01 + s_ie^{(t_iP_{TU_i})}\right] \tag{27}$$

In the past decade, the aggravated global warming and air pollution has driven many countries to commit to reducing carbon emissions [46]. The carbon tax ($C_{tax}$) emerged in order to penalize greenhouse gas emissions and encourage enterprise to invest renewable energy. Therefore, the cost of emission can be expressed as:

$$C_E = C_{tax} \times E \tag{28}$$

### 2.2.5. Objective Functions

Similar to conventional EED problems, the objective of this mathematical model is to optimize the output of each generating unit to meet the load demand and simultaneously reduce the generation cost and emission as much as possible under the condition of satisfying various constraints. The difference lies in that decision variables here consist of the produced actual power and generator bus voltages. The first objective function is formulated incorporating the cost of conventional thermal units, direct cost, reserve cost, and penalty cost as mentioned above. The second objective function principally takes environment factors into account and aims to minimize the total emission of conventional thermal generators. Moreover, the carbon tax is also incorporated into the objective function as discussed above. Thus, all the objective functions can be expressed as follows:

$$
\begin{aligned}
F_1 = C_{To}(P_{TU}) + \sum_{i=1}^{N_{WG}} & \left[ C_{\text{wind},i}(P_{sw,i}) + C_{RC,i}(P_{sw,i} - P_{wa,i}) + C_{PC,i}(P_{wa,i} - P_{sw,i}) \right] \\
+ & \sum_{j=1}^{N} \left[ C_{\text{solar},j}(P_{ss,j}) + C_{SR,j}(P_{ss,j} - P_{sa,j}) + C_{PS,j}(P_{sa,j} - P_{ss,j}) \right]
\end{aligned}
\tag{29}
$$

$$F_2 = C_E = C_{tax} \times E \tag{30}$$

### 2.2.6. Constraints

There are equality and inequality constraints that must be met. Equality constraints refers to the balance between active and reactive power generated with load demand and transmission losses. Hence, equality constraints are formulated as follows:

$$P_{Gi} - P_{Di} - V_i \sum_{j=1}^{NB} V_j \left[ G_{ij} \cos(\delta_{ij}) + B_{ij} \sin(\delta_{ij}) \right] = 0 \tag{31}$$

$$Q_{Gi} - Q_{Di} - V_i \sum_{j=1}^{NB} V_j \left[ G_{ij} \sin(\delta_{ij}) - B_{ij} \cos(\delta_{ij}) \right] = 0 \tag{32}$$

The inequality constraints cover the operation limits on equipment and security constraints on lines and load buses. Generator constraints:

$$P_{TU_i}^{min} \leq P_{TU_i} \leq P_{TU_i}^{max}, i = 1, 2, \ldots, N_{TU} \tag{33}$$

$$Q_{TU_i}^{min} \leq Q_{TU_i} \leq Q_{TU_i}^{max}, i = 1, 2, \ldots, N_{TU} \tag{34}$$

$$P_{sw,j}^{min} \leq P_{sw,j} \leq P_{sw,j}^{max}, j = 1, 2, \ldots, N_{WG} \tag{35}$$

$$Q_{sw,j}^{min} \leq Q_{sw,j} \leq Q_{sw,j}^{max}, j = 1, 2, \ldots, N_{WG} \tag{36}$$

$$P_{ss,k}^{min} \leq P_{ss,k} \leq P_{ss,k}^{max}, k = 1, 2, \ldots, N_{SG} \tag{37}$$

$$Q_{ss,k}^{min} \leq Q_{ss,k} \leq Q_{ss,k}^{max}, k = 1, 2, \ldots, N_{SG} \tag{38}$$

$$V_{G_i}^{min} \leq V_{G_i} \leq V_{G_i}^{max}, i = 1, 2, \ldots, N_G \tag{39}$$

Security constraints:

$$V_{L_p}^{min} \leq V_{L_p} \leq V_{L_p}^{max}, i = 1, 2, \ldots, NL \tag{40}$$

$$S_{l_q} \leq S_{l_q}^{max}, q = 1, 2, \ldots, nl \tag{41}$$

Based on the above formula, Equations (33) and (34) represent the limitation of the active and reactive power generated by thermal generators. Equations (35) and (36) are the limitations of the active and reactive power generated by wind generators. Equations (37) and (38) formulate the limitation of the active and reactive power generated by solar PV. Equation (39) represents the constraints on voltage of generator buses. Equation (40) represents the limitation of voltage imposed on *NL* numbers of load buses. Equation (41) denotes the line capacity constraints with total nl numbers of lines in the network.

### 2.3. Combined Emission Economic Dispatch Problems with Wind Penetration

Analogous to the conventional EED problem, the combined emission economic dispatch problem with wind power penetration also aims to achieve optimal scheduling of power generators to minimize the fuel cost and emission generated by thermal generators while simultaneously satisfying all the equality and inequality constraints. The objective functions of this system are the same as defined in Equations (10) and (27). The only difference is that wind energy becomes a part of energy mix and supplies a portion of the power demand. Similarly to the CHEED model mentioned above, the wind energy is counted as negative loads. The power balance constraint is converted into:

$$\sum_{i=1}^{N} P_{T_i} + P_{wp} = P_D + P_{\text{Loss}} \tag{42}$$

where $N$ is the number of thermal generators and $P_{T_i}$ stands for the output power of i-th thermal generator. $P_{wp}$ and $P_D$ represent the outputs of the wind farm and the load demand of the system, respectively. $P_{\text{Loss}}$ is the transmission loss, which can be formulated by the B matrix coefficients:

$$P_{\text{Loss}} = \sum_{i=1}^{N} \sum_{j=1}^{N} P_{Ti} B_{ij} P_{Tj} + \sum_{i=1}^{N} B_{oi} P_{Ti} + B_{oo} \tag{43}$$

where $B_{ij}$, $B_{oi}$, and $B_{oo}$ are matrix coefficients of the transmission loss.

### 3. The Proposed Optimization Algorithm
#### 3.1. Overview of the Cross Entropy Method

As mentioned above, the cross entropy (CE) method was first proposed in 1997 by Robinstein [25,26] based on rare events resampling and the Kullback–Leibler distance minimization. The cross-entropy method was initially employed in mathematics for the calculation of low probability events and then extended to numerical optimization. It is an optimization method that updates new parameters by collecting information in each iteration. The implementation procedure of CE is briefly divided into two major parts: First, random samples are generated based on the probability density function (PDF) and obtain the fitness values according to objective functions. Second, the elite set is used to update the PDF and create new samples.

For simplicity, the critical process of the conventional CE method is presented here. It is assumed that every individual *i* solution has a D-dimensional position vector, which is formulated by $X_i^k = \left( x_1^k, x_2^k, \ldots, x_D^k \right)$ at iteration $k$. $\mu_n$, $\sigma_n$ are defined as the mean value and the standard deviation of the dimension $n$, respectively. The components of each individual solution are selected independently and specifically generated separately based on the $N\left( \mu_n, \sigma_n^2 \right)$ distribution. Then, the fitness values of all individuals are calculated and sorted in ascending order. For the minimization problems, $\rho \times N_p$ best individuals are chosen as the elite set $X^{elite}$ to help guide the updating process of the mean value and the standard deviation where $N_p$ is the population size and $\rho$ is a quantile. The population can be generated by Equation (44), where $x_n^j$ stands for the value of the j-th individual in the

n-th dimension and $randn()$ is a random variable based on normal distribution function $N(0,1)$. The evolution of an individual to a new individual is conducted in terms of the mean and standard deviation of the elite set. The updating process are normally given as follows:

$$x_n^j = \mu_n + \text{randn} () \times \sigma_n \tag{44}$$

$$X_n^{\text{elite}}(k) = \left(x_n^1, x_n^2, \ldots, x_n^{\rho \times N}\right) \tag{45}$$

$$\mu_n(k+1) = \text{mean}\left(X_n^{\text{elite}}(k)\right) \tag{46}$$

$$\sigma_n(k+1) = \text{std}\left(X_n^{\text{elite}}(k)\right) \tag{47}$$

However, it is far from being enough to merely rely on the elite set to update samples. Hence, smoothing parameters are introduced for the improvement of the searching capacity. In Equations (48) and (49), $\alpha$ and $\beta$ are defined as smoothing parameters. $\mu(k+1)$ and $\varsigma(k+1)$ are employed to produce new samples, hoping to discover better solutions. Here, $\alpha$ and $\beta$ are both constants (typically ranging from 0.8 and 0.99), corresponding to a fixed version of the CE method (FCE).

$$\mu(k+1) = \alpha\mu(k+1) + (1-\alpha)\mu(k) \tag{48}$$

$$\varsigma(k+1) = \beta\varsigma(k+1) + (1-\beta)\varsigma(k) \tag{49}$$

Moreover, it should be noted that FCE is able to show good performance in many cases but not in some other situations. Therefore, a dynamic version of the CE method (DCE) emerges. For $\mu$, the same fixed parameter described in Equation (48) is used; the dynamic adjustment of smoothing parameters for standard deviation is expressed by:

$$\beta_n = C\left(1 - \left(1 - \frac{1}{k}\right)^q\right) \tag{50}$$

where $\beta_n$ denotes the value of $\beta$ of the n-th decision variable in the k-th iteration. $C$ is a constant between 0.8 and 0.99, and $q$ is an integer between five and ten.

In the global search, the population is regenerated after the completion of the updating process of the mean value and the standard deviation. Then, the individuals in the new population are rearranged from largest to smallest according to fitness values and the elite set is selected from among them. The above steps are repeated for a specific number of iterations.

### 3.2. The Proposed Modified Multi-Objective Cross-Entropy Algorithm

Since most of the practical engineering optimization problems are multi-objective optimization problems, the CE method needs to be improved to adapt to such problems. In this paper, in order to enhance the adaptability of the algorithm in multi-objective optimization, the CE method is extended to a multi-objective algorithm due to the introduction of the framework of NSGA-II [50]. This method couples with a fast non-dominated sorting approach to choose the elite set and stores the Pareto solutions in the external archive. The modified multi-objective cross entropy algorithm (MMOCE) maintains the advantage of fast convergence. However, there is a risk of falling into local optimum in the process of global search, and the ability of the algorithm to solve large scale system is truly unsatisfactory. Hence, a crossover operator is integrated into the algorithm to increase the diversity of solutions and improve the evolutionary efficiency. Additionally, pre-determined parameters in the conventional CE method may influence the optimization results because of improper choice. Although DCE and FCE can be sometimes effective, the options of proper parameters are not so simple. To tackle this problem, a new version of the mean value and the standard deviation updating process is proposed. With no need for setting a constant parameter, the process of updating the mean value and the standard deviation

is self-adaptive. This mechanism simplifies the algorithm and simultaneously ensure the convergence. Moreover, different parameter evolutionary mechanisms as stated in [41] are employed to this algorithm, which enhance the diversity of solutions and speed up the convergence rate. Analogous to [41], in order to facilitate the balance between exploitation and exploration searches, the whole iterative process is divided into the diversification stage and intensification stage. The diversification stage highlights the diversity of sampling points, while the intensification stage guarantees the rapid convergence. More details of the different parameter evolutionary mechanisms can be found in [41]. The specific introduction of the self-adaptive parameter updating process and crossover mechanism is given as follows.

### 3.2.1. Updating of Self-Adaptive Parameter

As mentioned earlier, the updating process of the mean value and the standard deviation involves the smoothing parameter. These constant parameters are often pre-determined and application-dependent. To avoid this, a self-adaptive parameter updating process is introduced here, and the mean value and the standard deviation can be updated in a self-adaptive way. This self-adaptive updating process can be implemented as follows at iteration $k$:

$$\mu(k+1) = \mu(k+1) \tag{51}$$

$$\beta = 0.382 \times rand \tag{52}$$

$$\varsigma(k+1) = \beta\varsigma(k+1) + (1-\beta)\varsigma(k) \tag{53}$$

where *rand* represents a random value ranging from 0 and 1. Considering the widespread use of the self-adaptive thinking in real-life optimization, Equation (50) can be reformulated as Equation (52). From the listed formula above, we can see that the mean value and standard deviation are self-adaptive after initialization and avoid the trouble of setting parameters in advance. The test functions below can demonstrate the validity and effectiveness of the method.

### 3.2.2. Crossover Operator

To increase the diversity and expand the search range, the crossover operator is introduced. The genetic information of elite individuals in the external archive is introduced to guide the evolution. First, the random individuals $X_i$ and $X_j$ are chosen from the present population P and the external archive, respectively. Second, part of the code of the two individuals is exchanged, generating two new individuals. The above operation is repeated until a new population P new with $N_p$ individuals is generated, where $N_p$ is the size of the population. This operation can be described as follows:

$$x_i^{new} = \begin{cases} x_j^n, rand < P_c \\ x_i^n, else \end{cases}, x_j^{new} = \begin{cases} x_i^n, rand < P_c \\ x_j^n, else \end{cases} \tag{54}$$

where $x_i^{new}$ and $x_j^{new}$ are the corresponding n-dimension encoding after the crossover operation. *rand* is a random number between 0 and 1; $P_c$ is the probability of crossover.

Figure 5 presents the flowchart of the proposed algorithm.

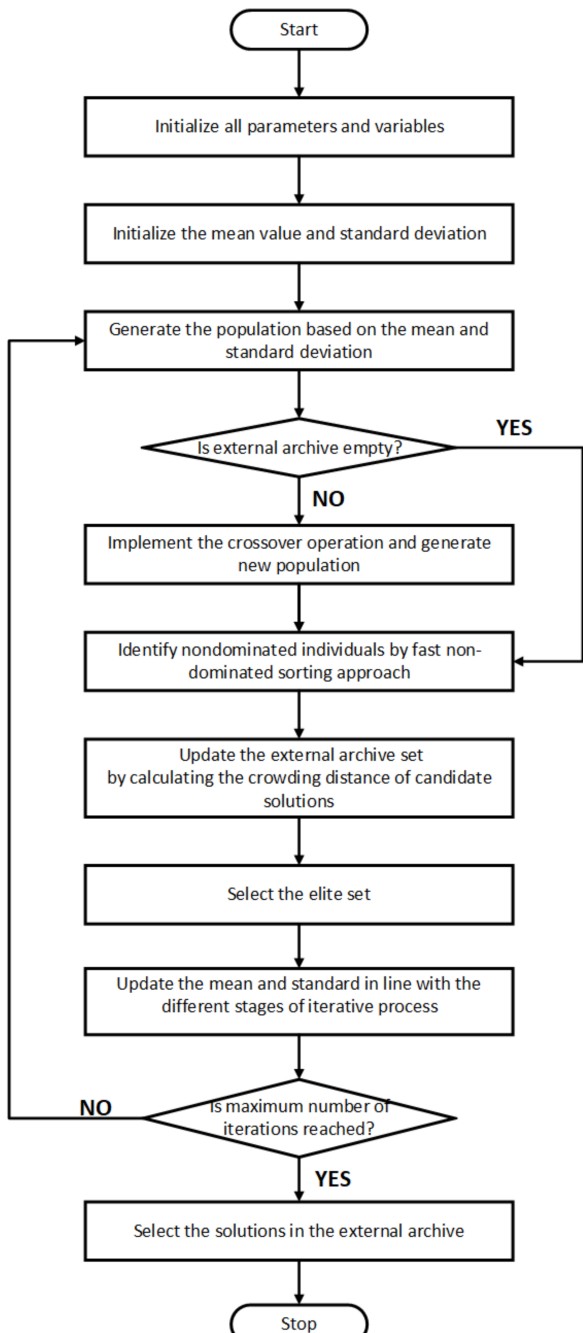

**Figure 5.** Flowchart of the proposed algorithm.

### 3.3. Implementation of MMOCE for EED

This subsection presents the proposed MMOCE procedure for solving the EED problems with renewable energy in detail.

- Step 1. Input data. The feasible range of control variables, the population size $N_p$, the maximum number of function evaluations Max_FES, the maximum size of external archive $Ar_{max}$, and the probability of crossover $P_c$.
- Step 2. Initialization. The initial population $P$ is formed by normal distribution. It is assumed that the dimension n over ranges $[X_{min}, X_{max}]$, which denote the lower and upper limit, is assigned based on $N(\mu_n, \sigma_n^2)$, where $\mu$ n is a random number between the lower and upper limit and $\varsigma$ n is set to $10 \cdot (X_{max} - X_{min})$, and FES = 0.

- Step 3. The mean value $\mu$ and standard deviation $\varsigma$ are updated by Equations (51)–(53). The endpoint value method is used as the boundary constraints handling strategy for the population $P$.
- Step 4. The objective function values are calculated and coupled with a fast non-dominated sorting approach and crowding–distance metric to get the external archive.
- Step 5. The crossover operator is carried out by Equation (54).
- Step 6. The elite set $X^{elite}$ is selected from the external archive and the different parameter evolutionary mechanisms are applied to update the mean value $\mu$ and standard deviation $\varsigma$. FES = FES + $N_p$.
- Step 7. Step 2 to Step 6 are repeated. If the termination criterion is met, i.e., Max_FES is reached, the above procedure is stopped.

## 4. Experimental Results

### 4.1. Experiments Based on Benchmark Functions

In order to verify the optimization performance of the proposed algorithm, eight medium–large scale test functions ZDT1-3, UF1-3, and WFG1-2 are selected. Meanwhile, five representative multi-objective evolutionary algorithms (NSGAII, MOEA/D, MOPSO, PESA2, MODE [51]) are also tested for comparison purposes. In addition, the DCE and FCE are both extended to multi-objective algorithms, denoted as DMOCE and FMOCE. The differences between the proposed algorithm and conventional CE variants are mainly reflected in the choice of smoothing parameters. Thus, four different multi-objective cross entropy optimization algorithms (DMOCE, FMOCE, MOO CEM [39], SMOCE [52]) are also implemented for comparison purposes. Moreover, this paper adopts the widely-used inverted generation distance and MaxSpread as performance indicators in multi-objective evolutionary algorithms. The inverted generation distance (IGD) calculates the Euclidean distance of the nearest Pareto optimal solution to the nearest Pareto optimal solution set, while MaxSpread (MS) is used to evaluate the coverage degree of the non-dominated solution obtained by the algorithm to Pareto theory optimal solution set. Their definitions are as follows:

4.1.1. Inverted Generation Distance (IGD)

$$GD = \frac{\sqrt{\sum_{i=1}^{|Q|} d_i^2}}{|Q|} \tag{55}$$

$$d_i = min\left\{ \sqrt{\sum \left( f_j^{(i)} - f_j^{*(k)} \right)^2} \right\}, k = 1, 2, \ldots, |Q^*| \tag{56}$$

where $Q$ denotes the Pareto optimal solution set obtained by the algorithm, $|Q|$ is the number of non-dominated solutions in $Q$, $Q^*$ denotes the theoretical Pareto optimal solution set, $|Q^*|$ denotes the number of non-dominated solutions, and $f_j^{(i)}$ and $f_j^{*(k)}$ represent the j-th objective function value of the i-th solution in Q and the j-th objective function value of the k-th solution, respectively.

4.1.2. MaxSpread (MS)

$$MS = \sqrt{\frac{1}{n} \sum_{i=1}^{n} \left( \frac{min(max(f_i), max(F_i)) - max(min(f_i), min(F_i))}{max(F_i) - min(F_i)} \right)^2} \tag{57}$$

where $f_i$ represents the value of the i-th objective function achieved by the algorithm corresponding to the solution and $F_i$ represents the objective value of the i-th objective of the theoretical optimal solution. If MS = 1, it means that the non-dominated solution found by the algorithm can completely cover the Pareto theoretical optimal solution.

In this paper, each algorithm runs 30 times independently to obtain the average values. The population size of each algorithm is set to 100, which is the same as [53–55], and the max function evaluations (FES) of each test function are 15,000 for all algorithms. The external archive set size is set to 100. The crossover probability of MMOCE is 0.9 and elite set size is set as 10. For NSGAII, the crossover probability is 0.9, the mutation probability is 0.1, and the mutation distribution index is 20. The mutation scale factors and crossover probability of MODE are 0.5 and 0.3, respectively. The probability of crossover and mutation of PESA2 is both 0.5 and the number of grids is 7. Finally, MOPSO uses the following parameters: the number of grids is set to 7, the probability of mutation is 0.1, both the individual and the global learning factors are 1 and 2 respectively, and the inertia weight is 0.5. All the simulations are run using MATLAB 2016a and on a 2.4 GHZ and 8 GM RAM computer in Windows 10.

Tables 1 and 2 are a summary of the influence of different constants in Equation (52) on the optimization performance of the proposed algorithm. IGD denotes the proximity between the non-dominated solution and the optimal Pareto front end, and MS can evaluate the coverage of the non-dominated solution to the optimal solution. It is evident that when the constant is 0.382, the proposed algorithm can achieve overall better optimization performance compared to other values. When the constant is 1.0, it is clear that the optimization performance of the algorithm deteriorates dramatically. The primary reason is the lack of perturbation, and the diversity of solutions decreases sharply and premature convergence occurs. Therefore, the diversity of solutions is of great significance to further improving the searching ability of the algorithm. Compared to other values, when the constant is set to 0.382, the eight test functions with different characteristics show better optimization performance.

**Table 1.** Comparison of influences on IGD indicators of various constant values.

|  | ZDT1 | ZDT2 | ZDT3 | UF1 | UF2 | UF3 | WFG1 | WFG2 |
|---|---|---|---|---|---|---|---|---|
| 0.1 | $3.23 \times 10^{-4}$ | $4.43 \times 10^{-4}$ | $4.02E \times 10^{-4}$ | **0.0051** | 0.0039 | 0.0226 | 0.0362 | 0.0257 |
| 0.2 | $2.64 \times 10^{-4}$ | $2.62 \times 10^{-4}$ | $2.89E \times 10^{-4}$ | 0.0051 | 0.0030 | **0.0222** | 0.0352 | 0.0258 |
| 0.3 | $2.58 \times 10^{-4}$ | $2.59 \times 10^{-4}$ | $2.90E \times 10^{-4}$ | 0.0052 | 0.0029 | 0.0225 | 0.0352 | 0.0257 |
| 0.382 | **$2.57 \times 10^{-4}$** | **$2.58 \times 10^{-4}$** | **$2.85 \times 10^{-4}$** | 0.0052 | **0.0029** | 0.0223 | 0.0351 | **0.0247** |
| 0.5 | $2.59 \times 10^{-4}$ | $2.58 \times 10^{-4}$ | $2.89 \times 10^{-4}$ | 0.0052 | 0.0030 | 0.0234 | 0.0349 | 0.0271 |
| 0.6 | $2.65 \times 10^{-4}$ | $2.65 \times 10^{-4}$ | $2.99 \times 10^{-4}$ | 0.0053 | 0.0032 | 0.0222 | 0.0348 | 0.0298 |
| 0.7 | $2.78 \times 10^{-4}$ | $2.87 \times 10^{-4}$ | $3.48 \times 10^{-4}$ | 0.0055 | 0.0033 | 0.0224 | 0.0348 | 0.0284 |
| 0.8 | $2.97 \times 10^{-4}$ | 0.0026 | $3.26 \times 10^{-4}$ | 0.0055 | 0.0033 | 0.0248 | **0.0346** | 0.0287 |
| 0.9 | $2.94 \times 10^{-4}$ | 0.0060 | 0.0010 | 0.0055 | 0.0035 | 0.0229 | 0.0347 | 0.0305 |
| 1.0 | 0.0013 | 0.0135 | 0.0017 | 0.0057 | 0.0040 | 0.0240 | 0.0346 | 0.0293 |

**Table 2.** Comparison of influences on MS indicators of various constant values.

|  | ZDT1 | ZDT2 | ZDT3 | UF1 | UF2 | UF3 | WFG1 | WFG2 |
|---|---|---|---|---|---|---|---|---|
| 0.1 | 1 | 1 | 0.9974 | 0.9955 | 0.9754 | 0.8170 | 0.6574 | **0.8072** |
| 0.2 | 1 | 1 | 0.9994 | 0.9987 | 0.9854 | 0.7510 | 0.6612 | 0.7899 |
| 0.3 | 1 | 1 | 0.9996 | 0.9991 | 0.9854 | 0.7518 | 0.6623 | 0.7814 |
| 0.382 | **1** | **1** | **0.9997** | **0.9994** | **0.9857** | **0.8602** | 0.6650 | 0.7800 |
| 0.5 | 0.9996 | 0.9998 | 0.9994 | 0.9991 | 0.9853 | 0.7160 | 0.6649 | 0.7493 |
| 0.6 | 0.9999 | 0.9990 | 0.9986 | 0.9985 | 0.9856 | 0.7351 | 0.6651 | 0.7463 |
| 0.7 | 0.9993 | 0.9979 | 0.9978 | 0.9958 | 0.9832 | 0.5762 | 0.6654 | 0.7227 |
| 0.8 | 0.9990 | 0.9438 | 0.9979 | 0.9930 | 0.9817 | 0.6401 | 0.6658 | 0.6888 |
| 0.9 | 0.9986 | 0.8798 | 0.9914 | 0.9957 | 0.9761 | 0.5692 | 0.6649 | 0.6783 |
| 1.0 | 0.9866 | 0.8222 | 0.9845 | 0.9934 | 0.9570 | 0.5633 | **0.6669** | 0.6730 |

Tables 3 and 4 present the IGD data of MMOCE and other algorithms. It can be observed from the tables that the MMOCE has better IGD characteristics than other algorithms for the function ZDT1-3 and WFG1-2, and it has the second best IGD for UF1-2

after MODE. Although not as good as other algorithms for UF3, it is obvious that the proposed algorithm is generally better than other algorithms on the IGD index. It implies that the non-dominated solutions obtained by the proposed algorithm is the closest to the theoretical optimal frontier.

**Table 3.** Comparison of various evolutionary algorithms IGD indicators.

|  | MODE | MOEA/D | MOPSO | NSGAII | PESAII | MMOCE |
|---|---|---|---|---|---|---|
| ZDT1 | 0.0065 | 0.0072 | 0.0021 | 0.0263 | 0.0047 | $\mathbf{2.57 \times 10^{-4}}$ |
| ZDT2 | 0.0590 | 0.0569 | 0.0305 | 0.0524 | 0.0071 | $\mathbf{2.58 \times 10^{-4}}$ |
| ZDT3 | 0.0215 | 0.0157 | 0.0051 | 0.0322 | 0.0056 | $\mathbf{2.85 \times 10^{-4}}$ |
| UF1 | **0.0048** | 0.0178 | 0.0169 | 0.0178 | 0.0082 | 0.0052 |
| UF2 | **0.0024** | 0.0065 | 0.0099 | 0.0065 | 0.0059 | 0.0030 |
| UF3 | **0.0174** | 0.0225 | 0.0273 | 0.0271 | 0.0214 | 0.0223 |
| WFG1 | 0.0722 | 0.0352 | 0.0376 | 0.0440 | 0.0372 | **0.0351** |
| WFG2 | 0.0308 | 0.0540 | 0.0473 | 0.0529 | 0.0365 | **0.0247** |

**Table 4.** Comparison of various multi-objective cross entropy algorithms IGD indicators.

|  | FMOCE | DMOCE | MOO CEM | SMOCE | MMOCE |
|---|---|---|---|---|---|
| ZDT1 | 0.0031 | 3.4564E-04 | 0.0109 | 0.0192 | $\mathbf{2.5709 \times 10^{-4}}$ |
| ZDT2 | 0.0269 | 8.5574E-04 | 0.0015 | 0.0462 | $\mathbf{2.5821 \times 10^{-4}}$ |
| ZDT3 | 0.0046 | 4.6270E-04 | 0.0154 | 0.0162 | $\mathbf{2.8454 \times 10^{-4}}$ |
| UF1 | 0.0058 | 0.0053 | 0.0079 | 0.0077 | **0.0052** |
| UF2 | 0.0043 | 0.0030 | 0.0072 | 0.0046 | **0.0029** |
| UF3 | 0.0243 | 0.0211 | **0.0137** | 0.0186 | 0.0223 |
| WFG1 | 0.0358 | 0.0353 | 0.0937 | 0.0803 | **0.0351** |
| WFG2 | 0.0301 | 0.0289 | 0.0635 | 0.0452 | **0.0247** |

Tables 5 and 6 compare the performance of MMO CE algorithm and the MS index with other algorithms. It can be seen from the tables that the MMOCE outperforms other algorithm on the MS index, which implies that the proposed algorithm has the maximum coverage of the non-dominated solution to the optimal solution. Meanwhile, it indicates that the MMOCE has the best diversity among all the algorithm. Additionally, Table 7 lists the simulation time of different algorithms on benchmark functions. It is obvious that MMOCE has certain advantages that benefit from its simple structure.

**Table 5.** Comparison of various evolutionary algorithms MS indicators.

|  | MODE | MOEA/D | MOPSO | NSGAII | PESAII | MMOCE |
|---|---|---|---|---|---|---|
| ZDT1 | 0.9091 | 0.8825 | 0.9830 | 0.7267 | 0.9066 | **0.9999** |
| ZDT2 | 0.7507 | 0.9269 | 0.9583 | 0.5724 | 0.8386 | **1.0000** |
| ZDT3 | 0.9264 | 0.7465 | 0.9414 | 0.6296 | 0.9429 | **0.9996** |
| UF1 | 0.8108 | 0.3753 | 0.8954 | 0.7695 | 0.7783 | **0.9994** |
| UF2 | 0.9398 | 0.6874 | 0.8911 | 0.9092 | 0.8606 | **0.9837** |
| UF3 | 0.5529 | 0.1912 | 0.4928 | 0.5289 | 0.4153 | **0.8602** |
| WFG1 | 0.1796 | 0.5308 | 0.5819 | 0.3302 | 0.6397 | **0.6650** |
| WFG2 | 0.6554 | 0.4462 | 0.5437 | 0.4455 | 0.6282 | **0.7800** |

**Table 6.** Comparison of various multi-objective cross entropy algorithms MS indicators.

|        | FMOCE  | DMOCE  | MOO CEM | SMOCE  | MMOCE  |
|--------|--------|--------|---------|--------|--------|
| ZDT1   | 0.9947 | 0.9989 | 0.9712  | 0.7461 | **0.9999** |
| ZDT2   | 0.8573 | 0.9811 | 0.9693  | 0.7446 | **1.0000** |
| ZDT3   | 0.9680 | 0.9972 | 0.9350  | 0.8211 | **0.9996** |
| UF1    | 0.9909 | 0.9990 | 0.7454  | 0.8489 | **0.9994** |
| UF2    | 0.9464 | 0.9724 | 0.9349  | 0.8679 | **0.9837** |
| UF3    | 0.4954 | 0.5511 | 0.7204  | 0.5759 | **0.8602** |
| WFG1   | 0.6631 | 0.6621 | 0.2891  | 0.2589 | **0.6650** |
| WFG2   | 0.7146 | 0.6942 | 0.4726  | 0.5242 | **0.7800** |

**Table 7.** Comparison of various multi-objective cross entropy algorithms on average simulation time.

|        | MODE   | MOEA/D  | MOPSO  | NSGAII  | PESAII  | MMOCE  |
|--------|--------|---------|--------|---------|---------|--------|
| ZDT1   | 32.94  | 124.91  | 25.95  | 2.85    | 52.72   | **2.69** |
| ZDT2   | 31.67  | 124.56  | 4.16   | **5.71**| 42.26   | 3.08   |
| ZDT3   | 31.75  | 127.96  | 8.19   | **3.78**| 44.60   | 5.28   |
| UF1    | 31.67  | 128.73  | 6.70   | **6.02**| 29.42   | 10.38  |
| UF2    | 44.40  | 130.96  | 9.69   | **3.69**| 55.84   | 5.93   |
| UF3    | 40.87  | 139.01  | 18.26  | **3.79**| 50.44   | 11.19  |
| WFG1   | 55.59  | 131.20  | 20.48  | 4.46    | 70.60   | **3.21** |
| WFG2   | 52.88  | 126.48  | 6.66   | **4.53**| 41.68   | 8.19   |

*4.2. Simulation Results on Combined Heat, Emission, and Economic Dispatch (CHEED)*

This section discusses the implementation of the proposed MMOCE algorithm on the combined heat, emission, and economic dispatch problem in [56]. In this work, it is assumed that wind and solar energy each account for 5% of the total load requirements in all test cases. In this section, two power systems are used to examine the superiority of this proposed algorithm in solving the CHEED problems considering renewable energy sources. The first power system is composed of four conventional thermal units, two cogeneration units, and one heat-only unit. $P_d$ and $H_d$ are set to 600 and 150 $MW_{th}$, respectively. Note that this power system considers the power losses. The second power system consists of one conventional thermal unit, three cogeneration units, and one heat only unit, and two scenarios are considered. In the first scenario $P_d$ and $H_d$ are set to 300 $MW_{th}$ and 150 $MW_{th}$, while in the second scenario, $P_d$ and $H_d$ are set to 250 MW and 175 $MW_{th}$, respectively. The detail data of both power systems can be referred to in [42].

In order to confirm the feasibility and effectiveness of the proposed algorithm, the simulated results are employed to compare with one newly proposed MSFLA [56]. For coping with the given optimization model, the MSFLA introduces the price penalty factor (PPF) to combine the total cost and total emission into a single objective optimization model. The PPF in this paper is set to 35,863.1065 $/kg in the first system and 28,297.8499 $/kg in the second system. Thus, for convenient comparison, the best solution of the MMOCE (termed as $S_n$) is selected from the Pareto front according to $\min F = C_t + \Lambda E_t$ [56]. The maximum number of iterations is 100, and the FES for all the algorithms is set to 5000. Tables 8–10 list the detailed results of the two power systems derived from MMOCE. It can be observed that these results obtained by MMOCE are clearly more environmental and cost-efficient, which demonstrates the effectiveness and superiority of the method over MSFLA [56], GA [57], SFLA [58], TLBO [59], and ISFLA [60]. For the first system, the fuel cost and emission are 16,286.4264 $/ and 5.0793 $/h, respectively, by using MMOCE, and the total cost is 135178.144 $/h compared with the minimum total cost in MSFLA, which gives 154,574.6139 $/h. Similarly for the second system, the proposed MMOCE obtains the great reduction in total cost over GA, SFLA, TLBO, ISFLA, and MSFLA with 58,355.0346 $/h, 57,025.0000 $/h, 56,940.3826 $/h, 56,847.1008, $/h and 56,761.0001 $/h, respectively, for Scenario 1 of the second power system and 45,110.0000 $/h, 44,691.5777 $/h, 44,665.9095 $/h, 44,552.4498 $/h, and 44,393.269 6 $/h, respectively, for Scenario 2 of the

second power system. There is no doubt that in terms of fuel cost, emission, and total cost, the proposed algorithm enjoys a significant edge.

**Table 8.** Results of various methods for the first system of the first EED model.

| | Method | | | | | |
|---|---|---|---|---|---|---|
| | **GA [57]** | **SFLA [58]** | **TLBO [59]** | **ISFLA [60]** | **MSFLA [56]** | **MMOCE** |
| $P_1$ | 38.8755 | 48.3207 | 40.0649 | 39.3109 | 40.3650 | 35.6071 |
| $P_2$ | 53.7749 | 38.3751 | 51.0577 | 51.0506 | 51.0577 | 31.4136 |
| $P_3$ | 63.1298 | 61.5122 | 51.7601 | 51.9601 | 50.6388 | 47.6949 |
| $P_4$ | 58.3387 | 65.9934 | 72.4301 | 72.9411 | 72.0309 | 77.4051 |
| $P_5$ | 219.8752 | 219.8752 | 217.2403 | 217.4305 | 219.8752 | 245.8501 |
| $H_5$ | 75.9912 | 75.9912 | 74.9801 | 74.9731 | 75.9912 | 0.0000 |
| $P_6$ | 112.5602 | 112.5602 | 113.9616 | 113.8016 | 112.5602 | 108.4909 |
| $H_6$ | 69.4130 | 69.4130 | 69.1130 | 69.1030 | 69.4130 | 71.1467 |
| $H_7$ | 4.5957 | 4.5957 | 5.9071 | 5.9244 | 4.5957 | 78.8533 |
| $P_L$ | 6.5532 | 6.5440 | 6.5104 | 6.4942 | 6.4926 | 6.4818 |
| Fuel Cost | 16,345.2433 | 16,351.3678 | 16,314.2368 | 16,312.2403 | 16,344.6539 | **16,286.4264** |
| Emission | 6.0373 | 6.0021 | 5.9616 | 5.9572 | 5.9054 | **5.0793** |
| Total Cost | 157,660.9862 | 156,844.7707 | 155,859.8997 | 155,752.9672 | 154,574.6139 | **135,178.7443** |

**Table 9.** Results of various methods for Scenario 1 in the second system of the first EED model.

| | Method | | | | | |
|---|---|---|---|---|---|---|
| | **GA [57]** | **SFLA [58]** | **TLBO [59]** | **ISFLA [60]** | **MSFLA [56]** | **MMOCE** |
| $P_1$ | 36.2567 | 35.0000 | 35.1000 | 35.0400 | 35.0000 | 35.0327 |
| $P_2$ | 120.7361 | 105.7361 | 104.1568 | 104.3659 | 105.7361 | 107.2488 |
| $H_2$ | 65.5581 | 80.0100 | 91.7052 | 91.8122 | 92.0100 | 99.4770 |
| $P_3$ | 24.6440 | 47.7440 | 44.2212 | 44.0821 | 42.7440 | 22.7183 |
| $H_3$ | 44.0943 | 42.0943 | 42.3921 | 42.3844 | 42.0943 | 50.5230 |
| $P_4$ | 88.3632 | 81.5200 | 86.5220 | 86.5120 | 86.5200 | 105.0000 |
| $H_4$ | 40.3476 | 27.9037 | 15.9027 | 15.8034 | 15.9037 | 0.0000 |
| $H_5$ | 0.0000 | 0.0000 | 0.0000 | 0.0000 | 0.0000 | 0.0000 |
| Fuel Cost | 15,891.4751 | 15,955.0000 | 15,883.9895 | 15,886.4598 | 15,889.0000 | **15,716.8701** |
| Emission | 1.1840 | 1.1452 | 1.1448 | 1.1421 | 1.1397 | **1.1199** |
| Total Cost | 58,355.0346 | 57,025.0000 | 56,940.3826 | 56,847.1008 | 56,761.0001 | **55,879.9631** |

**Table 10.** Results of various methods for Scenario 2 in the second system of the first EED model.

| | Method | | | | | |
|---|---|---|---|---|---|---|
| | **GA [57]** | **SFLA [58]** | **TLBO [59]** | **ISFLA [60]** | **MSFLA [56]** | **MMOCE** |
| $P_1$ | 35.0356 | 35.0356 | 35.1688 | 35.2656 | 35.0356 | 35.0000 |
| $P_2$ | 95.0011 | 112.0611 | 112.5303 | 112.5013 | 112.0611 | 87.4635 |
| $H_2$ | 101.8674 | 107.8674 | 107.5905 | 107.8003 | 107.8674 | 112.6999 |
| $P_3$ | 38.4637 | 17.8637 | 10.4471 | 10.3338 | 10.4640 | 10.7293 |
| $H_3$ | 42.8607 | 38.8607 | 38.8596 | 38.8596 | 38.8607 | 40.3126 |
| $P_4$ | 56.5000 | 59.5000 | 66.8542 | 66.8993 | 66.8993 | 91.8075 |
| $H_4$ | 28.2738 | 28.2738 | 28.5500 | 28.3401 | 28.2738 | 21.9875 |
| $H_5$ | 2.0000 | 0.0002 | 0.0000 | 0.0000 | 0.0002 | 0.0000 |
| Fuel Cost | 14,421.0000 | 14,464.6435 | 14,389.7481 | 14,392.9789 | 14,397.6852 | **14,064.4171** |
| Emission | 1.0845 | 1.0681 | 1.0699 | 1.0658 | 1.0600 | **1.0449** |
| Total Cost | 45,110.0000 | 44,691.5777 | 44,665.9095 | 44,552.4498 | 44,393.2696 | **43,632.8405** |

Moreover, to further verify the effectiveness and superiority of the proposed MMOCE, three other multi-objective cross entropy (DMOCE, FMOCE, and MOO CEM) methods and two classic meta-heuristic multi-objective algorithms (NSGAII and MOPSO) are also used for comparison. Figure 6 show the Pareto optimal solutions of other algorithms in contrast

to the proposed method, respectively, in the first system. There are two explicit points: first, the Pareto optimal front is distinctly superior to other several multi-objective algorithms. Second, the Pareto solutions obtained by MMOCE are more evenly distributed, and the diversity is also excellent. Table 11 gives all the compromise solutions of all the algorithms in comparison. The solutions obtained by MMOCE acquire 12,714.9326 $/h in fuel cost and 10.1900 Kg/h in emission, which is reduced at least by 139.2994 $/h and 0.0644 Kg/h compared with other algorithms. Similarly, Figures 7 and 8 show the Pareto optimal solutions of different algorithms, and Tables 12 and 13 present all the compromise solutions of two scenarios in the second system of all the algorithms in comparison. Additionally, Figure 9 shows the comparison between the proposed MMOCE and other algorithms regrading to the total cost and emission for all scenarios. It can be clearly seen that overall the solutions obtained by MMOCE are better in convergence and diversity than those of other algorithms.

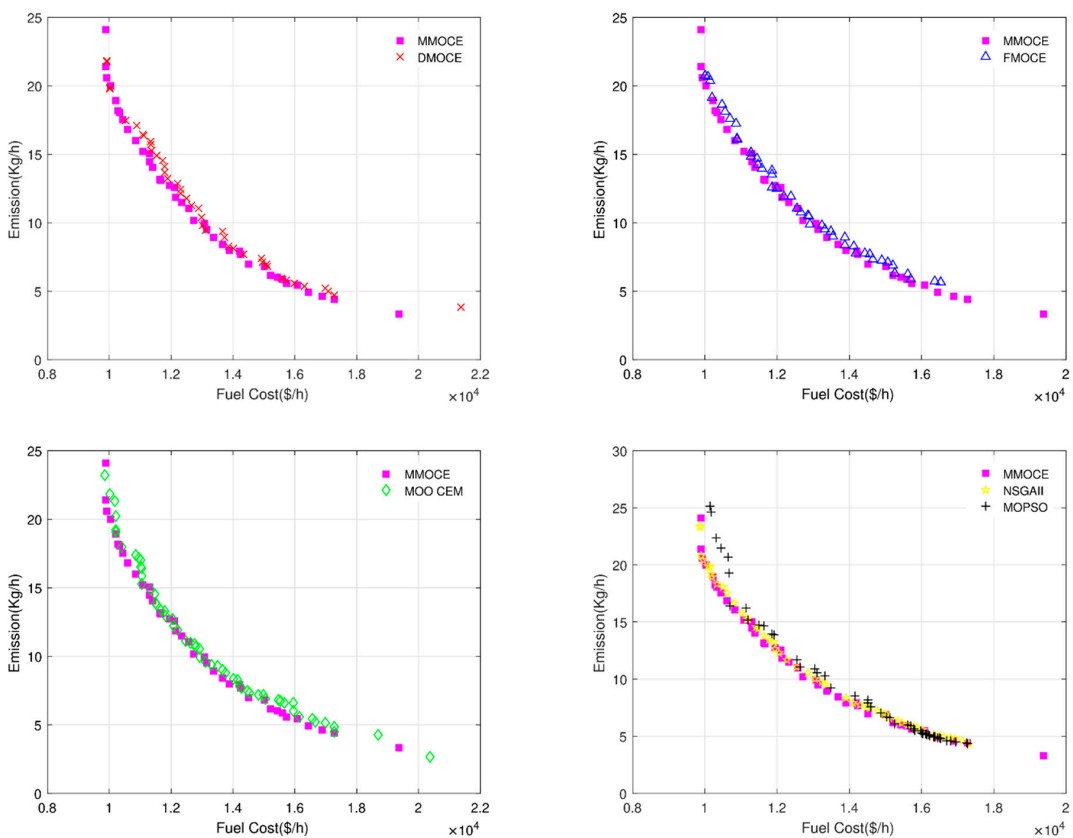

**Figure 6.** The Pareto front of MMOCE and other algorithms in the first system of the first EED model.

**Table 11.** Comparison of the compromise solutions obtained by all algorithms in the first system of the first EED model.

| Algorithms | Fuel Cost ($/h) | Emission (Kg/h) |
|:---:|:---:|:---:|
| **MMOCE** | **12,714.9326** | **10.1900** |
| FMOCE | 12,854.2320 | 10.5354 |
| DMOCE | 12,979.9436 | 10.3582 |
| MOO CEM | 12,929.2931 | 10.5643 |
| NSGAII | 12,855.0487 | 10.5610 |
| MOPSO | 13,319.9367 | 10.2544 |

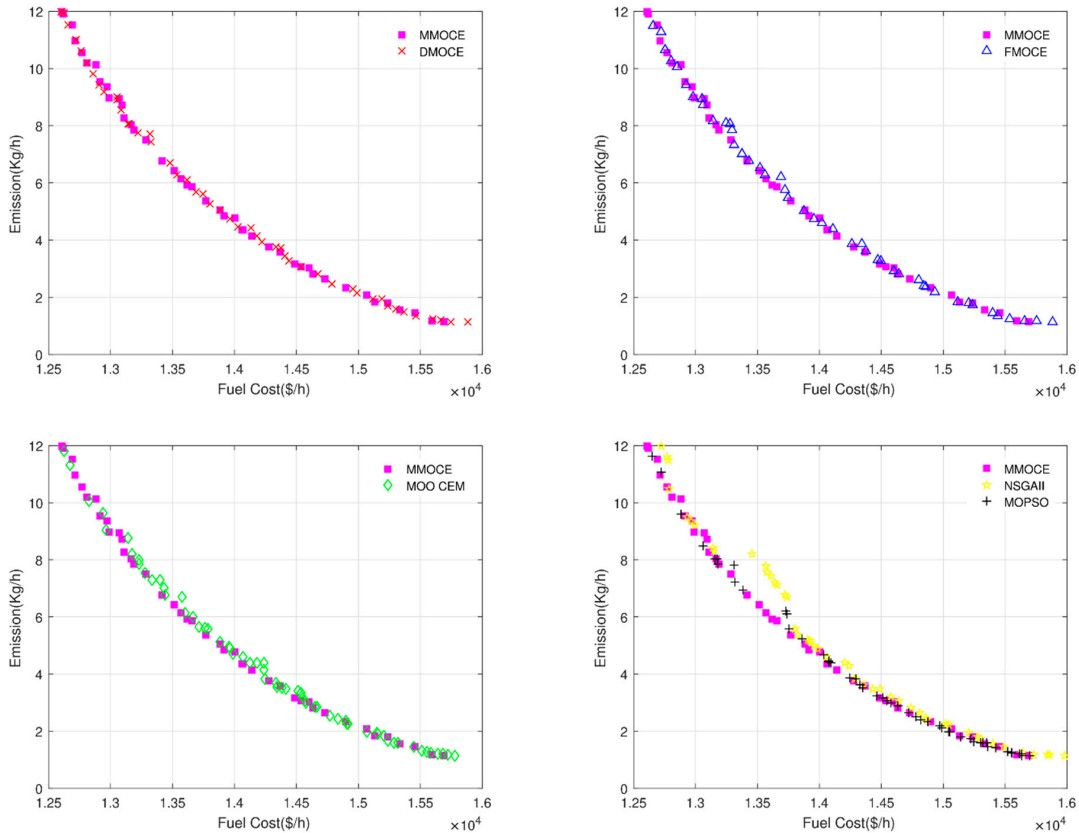

**Figure 7.** The Pareto front of MMOCE and other algorithms for Scenario 1 in the second system of the first EED model.

**Table 12.** Comparison of the compromise solutions obtained by all algorithms for Scenario 1 in the second system of the first EED model.

| Algorithms | Fuel Cost ($/h) | Emission (Kg/h) |
|:---:|:---:|:---:|
| MMOCE | **14,059.9237** | **4.3645** |
| FMOCE | 14,106.7250 | 4.3783 |
| DMOCE | 14,129.1889 | 4.4418 |
| MOO CEM | 14,205.7126 | 4.3789 |
| NSGAII | 14,099.5992 | 4.4725 |
| MOPSO | 14,089.1572 | 4.3871 |

**Table 13.** Comparison of the compromise solutions obtained by all algorithms for Scenario 2 in the second system of the first EED model.

| Algorithms | Fuel Cost ($/h) | Emission (Kg/h) |
|:---:|:---:|:---:|
| MMOCE | **12,252.9253** | **5.1305** |
| FMOCE | 12,267.3198 | 5.4654 |
| DMOCE | 12,278.0982 | 5.3153 |
| MOO CEM | 12,255.3025 | 5.2858 |
| NSGAII | 12,368.6973 | 5.2289 |
| MOPSO | 12,254.4826 | 5.1332 |

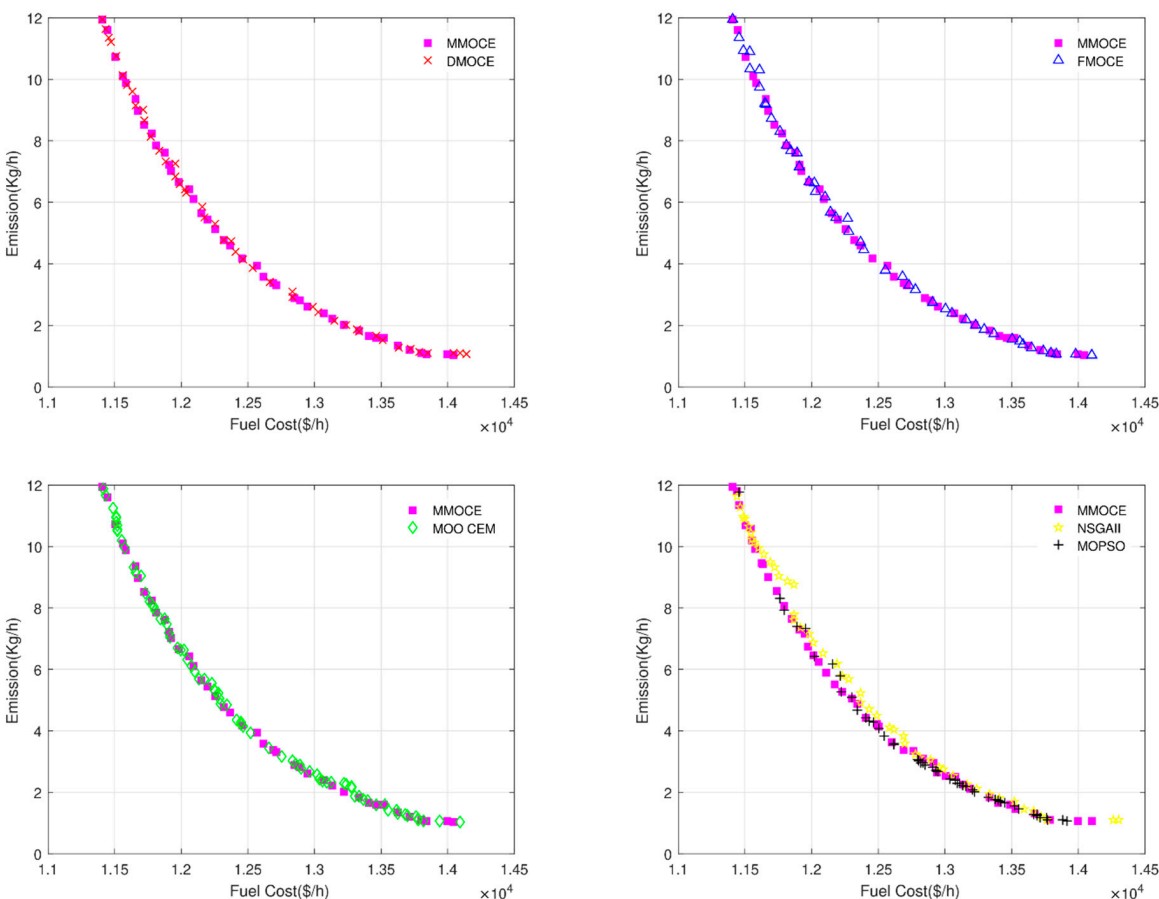

**Figure 8.** The Pareto front of MMOCE and other algorithms for Scenario 2 in the second system of the first EED model.

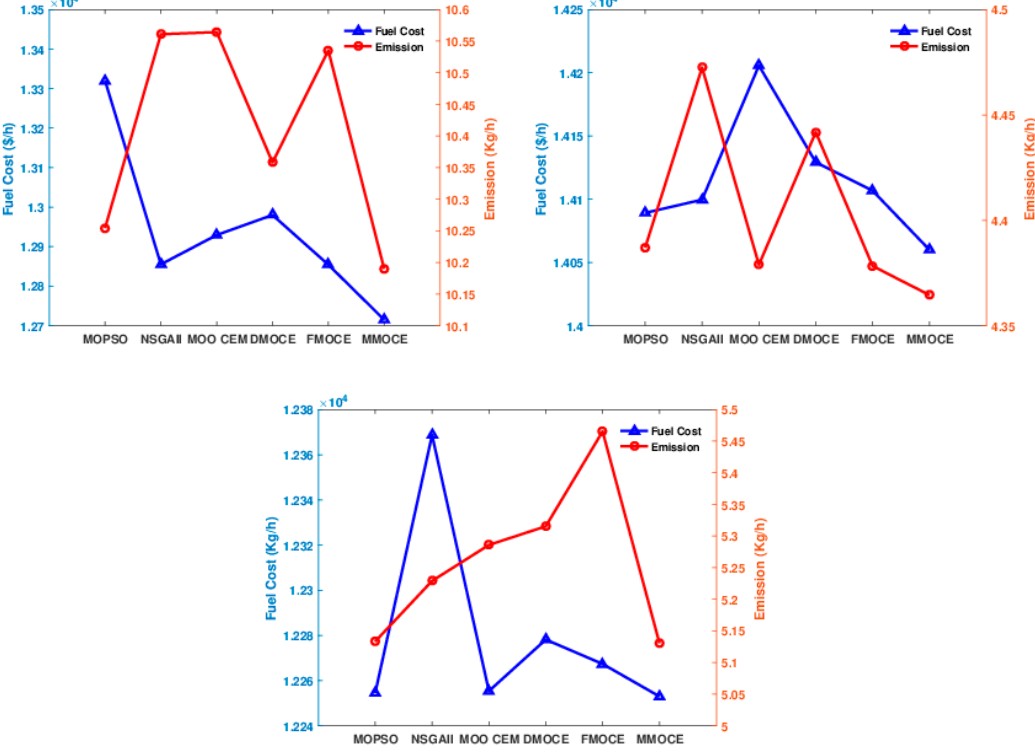

**Figure 9.** Total cost and emission for all methods in all systems of the first EED model.

### 4.3. Simulation Results on IEEE 30-Bus with Six-Generator System

In this section, three different cases of IEEE 30-bus with a six-generator system with renewable resources [61] are introduced, aiming to investigate the performance of the proposed MMOCE. In Case 1, the system incorporates stochastic wind and solar power. The Case 2 is a solar-free stochastic wind power system, while the third system is solar-only. Simultaneously, in order to verify the feasibility and validity of the proposed MMOCE, the CMOPEO-EED [61] method and CNSGAII-EED [50,62] method are selected as the competitors to participate in the optimization of three cases. Especially in Case 1, the results are also used to make comparison with that of the SHADE method in the literature [45]. The details of the three different cases of IEEE 30-bus with a six-generator system, renewable resources, and adjustable parameters of CMOPEO-EED and CNSGAII-EED can be found in [61]. The population size of the MMOCE is 60, and the maximum number of function evaluations is set to 24000, which is the same as that in the literature.

### 4.3.1. Case 1: System Incorporating Stochastic Wind and Solar

In this case study, the IEEE 30-bus system contains two wind farms and one solar PV array to generate electricity. In other words, this system considers the influence of stochastic wind and solar power at the same time. Firstly, the SHADE is used to make a comparison with the proposed MMOCE, as it treats the problem as a single-objective optimization problem. In order to highlight the outperformance of the MMOCE in contrast to the SHADE, the three optimal solutions (termed as $S_1 \sim S_3$) are extracted from the MMOCE on the basis of $F = F_1 + C_{tax}F_2$. Table 14 lists the results of the MMOCE and the SHADE in detail, and the possible range of each unit is also included. It is evident that the two Pareto solutions (i.e., $S_1$, $S_3$) obtained by the MMOCE are found to be superior to the optimal solution of the SHADE. $S_1$ has a total cost of 809.060 \$/h and an emission of 0.833 t/h, $S_3$ has a total cost of 809.655 \$/h and an emission of 0.744 t/h, while the SHADE has a total cost of 810.346 \$/h and an emission of 0.891 t/h. It is not difficult to discover that the proposed MMOCE yields lower total cost and emission than SHADE. The Pareto front of MMOCE is shown in Figure 10.

**Table 14.** Simulation results of three solutions from MMOCE and solution of SHADE in Case 1.

| Control Variables | Min | Max | MMOCE | | | SHADE [45] |
|---|---|---|---|---|---|---|
| | | | $S_1$ | $S_1$ | $S_1$ | |
| $P_{TG_1}$ (MW) | 50 | 140 | 122.352 | 128.012 | 120.35 | 123.525 |
| $P_{TG_2}$ (MW) | 20 | 80 | 32.955 | 31.274 | 32.955 | 33.047 |
| $P_{TG_3}$ (MW) | 10 | 35 | 10 | 10 | 10.124 | 10 |
| $P_{ws_1}$ (MW) | 0 | 75 | 45.898 | 43.969 | 45.898 | 37.336 |
| $P_{ws_1}$ (MW) | 0 | 60 | 38.43 | 37.663 | 40.203 | 46.021 |
| $P_{ss}$ (MW) | 0 | 50 | 38.998 | 38.022 | 38.998 | 38.748 |
| $V_1$ (p.u.) | 0.95 | 1.10 | 1.067 | 1.067 | 1.067 | 1.071 |
| $V_2$ (p.u.) | 0.95 | 1.10 | 1.052 | 1.053 | 1.053 | 1.057 |
| $V_5$ (p.u.) | 0.95 | 1.10 | 1.035 | 1.032 | 1.035 | 1.036 |
| $V_8$ (p.u.) | 0.95 | 1.10 | 1.10 | 1.1 | 1.1 | 1.04 |
| $V_{11}$ (p.u.) | 0.95 | 1.10 | 1.10 | 1.1 | 1.1 | 1.099 |
| $V_{13}$ (p.u.) | 0.95 | 1.10 | 1.067 | 1.061 | 1.067 | 1.056 |
| $Q_{TG_1}$ (MW) | −20 | 150 | −0.492 | −4.186 | −7.608 | 2.678 |
| $Q_{TG_2}$ (MW) | −20 | 60 | 0.2381 | 9.35 | 8.031 | 12.319 |
| $Q_{TG_3}$ (MW) | −15 | 40 | 40 | 40 | 40 | 35.27 |
| $Q_{ws1}$ (MW) | −30 | 35 | 24.184 | 21.62 | 23.429 | 22.964 |
| $Q_{ws2}$ (MW) | −25 | 30 | 30 | 30 | 30 | 30 |
| $Q_{ss}$ (MW) | −20 | 25 | 21.871 | 19.797 | 21.859 | 17.779 |
| $C_{tax}$ (\$/h) | | | 17.83 | 17.83 | 17.83 | 17.83 |
| Cost (\$/h) | | | **794.231** | 788.802 | **796.386** | / |
| Emission (t/h) | | | **0.833** | 1.16 | **0.744** | 0.891 |
| Tatal Cost (\$/h) | | | **809.060** | 809.478 | **809.655** | 810.346 |

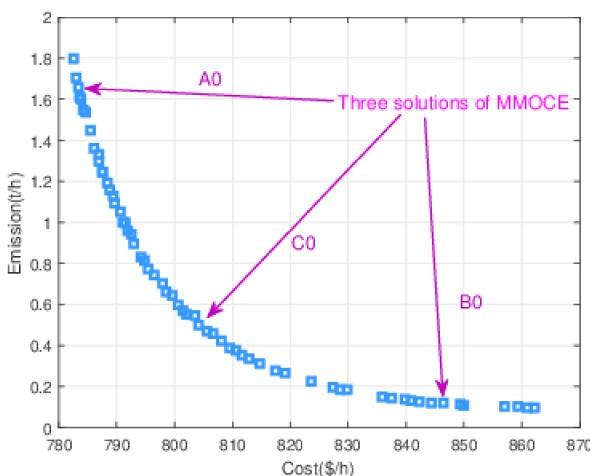

**Figure 10.** The Pareto front of MMOCE in Case 1.

In this paper, to compare with other two multi-objective algorithm CMOPEO-EED and CNSGAII-EED proposed in the literature, three different solutions (termed as A0, B0, and C0 in Figure 10) of the MMOCE are chosen from the Pareto front. As stated in [61], because of the preference to cost, A0 can be named cost solution. Similarly, B0 can be named emission solution, and C0 can be named as compromise solution. Table 15 lists the results of the three solutions in detail. It is shown that compared with other two algorithms, the result obtained from the proposed MMOCE is better as it achieves a decrease in both the cost and emission simultaneously in terms of cost solution and compromise solution. Though for emission solution, the MMOCE fails to show distinct superiority comparing with CMOPEO-EED, the proposed MMOCE still exhibits excellent convergence as a whole.

**Table 15.** Simulation results obtained by MMOCE, CMOPEO-EED, and CNSGAII-EED in Case 1.

| Control Variables | Cost-Solution (A0, A1, A2) | | | Emission-Solution (B0, B1, B2) | | | Compromise-Solution (C0, C1, C2) | | |
|---|---|---|---|---|---|---|---|---|---|
| | MMOCE | CMOEO-EED [61] | CNSGAII-EED [62] | MMOCE | CMOEO-EED [61] | CNSGAII-EED [62] | MMOCE | CMOEO-EED [61] | CNSGAII-EED [62] |
| $P_{TG_1}$ (MW) | 133.9071 | 134.2443 | 135.0588 | 69.1754 | 67.8734 | 72.1327 | 112.9751 | 113.6415 | 116.5484 |
| $P_{TG_2}$ (MW) | 26.9556 | 22.7365 | 24.0749 | 55.7616 | 52.4909 | 41.8570 | 37.0689 | 39.3716 | 36.2704 |
| $P_{TG_3}$ (MW) | 10 | 10.3261 | 16.8847 | 19.1664 | 14.3867 | 21.0405 | 10 | 10 | 16.6980 |
| $P_{ws_1}$ (MW) | 43.9685 | 44.4640 | 44.5520 | 54. 2363 | 54.7671 | 63.3223 | 49.9448 | 45.5167 | 45.0197 |
| $P_{ws_1}$ (MW) | 36.0219 | 39.3188 | 31.4980 | 46.6161 | 49.2786 | 46.7542 | 39.5683 | 38.7017 | 37.4236 |
| $P_{ss}$ (MW) | 38.2984 | 37.4988 | 37.4310 | 41.9088 | 48.0108 | 41.8268 | 38.6851 | 41.2587 | 36.9400 |
| $V_1$ (p.u.) | 1.0699 | 1.0749 | 1.0783 | 1.0665 | 1.0581 | 1.0783 | 1.0637 | 1.0581 | 1.0784 |
| $V_2$ (p.u.) | 1.0522 | 1.0606 | 0.9976 | 1.0413 | 1.0470 | 0.9917 | 1.0527 | 1.0516 | 0.9971 |
| $V_5$ (p.u.) | 1.0335 | 1.1000 | 1.0933 | 1.0323 | 1.1000 | 1.0933 | 1.0390 | 1.1000 | 1.0933 |
| $V_8$ (p.u.) | 1.1000 | 1.0453 | 1.1000 | 1.1000 | 1.0412 | 1.1000 | 1.0988 | 1.0341 | 1.0999 |
| $V_{11}$ (p.u.) | 1.0994 | 1.1000 | 1.0012 | 1.0994 | 1.1000 | 0.9868 | 1.0999 | 1.1000 | 0.9869 |
| $V_{13}$ (p.u.) | 1.0644 | 1.0084 | 1.1000 | 1.0567 | 1.1000 | 1.1000 | 1.0621 | 1.0637 | 1.0964 |
| $Q_{TG_1}$ (MW) | 2.1452 | −1.2440 | 35.9874 | 20.7012 | 1.5847 | 35.2648 | −7.8702 | −17.8574 | 38.7056 |
| $Q_{TG_2}$ (MW) | 0.3507 | 16.2791 | −20 | −20 | −14.1580 | −20 | 5.6642 | 16.9428 | −20 |
| $Q_{TG_3}$ (MW) | 40 | 40 | 40 | 40 | 40 | 40 | 40 | 40 | 40 |
| $Q_{ws1}$ (MW) | 23.7233 | 31.1651 | 35 | 21.7630 | 29.7936 | 35 | 26.6124 | 24.9220 | 35 |
| $Q_{ws2}$ (MW) | 29.8159 | 30 | 1.2666 | 30 | 30 | −3.6728 | 30 | 30 | −2.2033 |
| $Q_{ss}$ (MW) | 20.9158 | 1.4444 | 25 | 19.0388 | 25 | 25 | 20.0658 | 21.5430 | 25 |
| Cost ($/h) | **783.3389** | 784.0305 | 789.4062 | **846.5224** | 846.5317 | 848.2084 | **804.2061** | 804.5355 | 805.5781 |
| Emission | **1.6573** | 1.6931 | 1.7783 | 0.1168 | **0.1152** | 0.1215 | **0.4994** | 0.5167 | 0.6018 |

Furthermore, DMOCE, FMOCE, MOO CEM, NSGAII, and MOPSO are also tested for comparison purpose with the proposed MMOCE. Figure 11 compares the Pareto fronts between MMOCE and the above competitor algorithms. The compromise solutions of those algorithms are picked out and listed in Table 16. Apparently, the optimal solutions obtained by MMOCE unfold better convergence performance than MOO CEM, NSGAII,

and MOPSO. In comparison to FMOCE and DMOCE, the best Pareto fronts are visually close to each other. However, by means of the compromise solutions, it is clearly shown that MMOCE produces the cost of 804.2061 \$/h and emission of 0.4994 t/h, which is the lowest among these algorithms. In summary, the proposed algorithm MMOCE has demonstrated the superior performance in the optimization of this case.

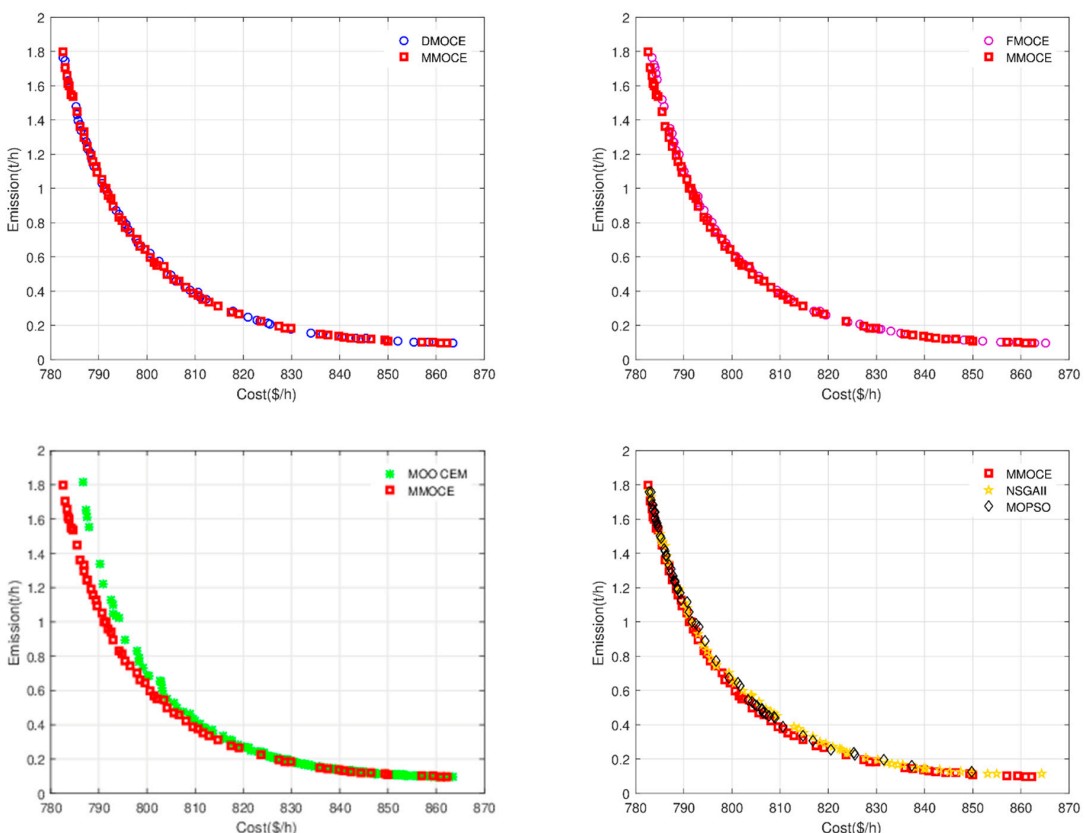

**Figure 11.** Pareto front of MMOCE and other algorithms in Case 1.

**Table 16.** Comparison of the compromise solutions obtained by all algorithms in Case 1.

|  | MMOCE | DMOCE | FMOCE | MOO CEM | NSGAII | MOPSO |
|---|---|---|---|---|---|---|
| $P_{TG_1}$ (MW) | 112.9751 | 113.0508 | 113.2180 | 113.9853 | 114.3649 | 113.5886 |
| $P_{TG_2}$ (MW) | 37.0689 | 38.9045 | 34.0862 | 30.6144 | 41.1093 | 40.9433 |
| $P_{TG_3}$ (MW) | 10 | 10 | 10 | 10.4010 | 11.1905 | 10.0105 |
| $P_{ws_1}$ (MW) | 49.9448 | 47.3610 | 50.4035 | 51.6109 | 42.2642 | 49.0710 |
| $P_{ws_1}$ (MW) | 39.5689 | 39.3637 | 41.8038 | 41.5601 | 43.9396 | 36.1924 |
| $P_{ss}$ (MW) | 36.6851 | 39.6871 | 38.7534 | 40.3542 | 35.7173 | 38.7900 |
| $V_1$ (p.u.) | 1.0637 | 1.0647 | 1.0735 | 1.0553 | 1.0646 | 1.0776 |
| $V_2$ (p.u.) | 1.0527 | 1.0509 | 0.9567 | 0.9702 | 0.9807 | 1.0267 |
| $V_5$ (p.u.) | 1.0390 | 1.0294 | 1.0350 | 1.0365 | 1.0856 | 1.0830 |
| $V_8$ (p.u.) | 1.0988 | 1.0361 | 1.0988 | 1.0323 | 1.0858 | 1.0952 |
| $V_{11}$ (p.u.) | 1.0999 | 1.1000 | 1.0848 | 1.0329 | 1.0999 | 1.0664 |
| $V_{13}$ (p.u.) | 1.0621 | 1.0490 | 1.0585 | 1.0243 | 1.0862 | 1.0198 |
| $Q_{TG_1}$ (MW) | −7.8702 | 0.2431 | 22.7539 | 27.1743 | 6.5293 | 29.8476 |
| $Q_{TG_2}$ (MW) | 5.6642 | 8.4721 | −20 | −20 | −20 | −20 |
| $Q_{TG_3}$ (MW) | 40 | 37.6685 | 40 | 40 | 40 | 40 |
| $Q_{ws1}$ (MW) | 26.6124 | 21.6926 | 25.7524 | 35 | 35 | 35 |
| $Q_{ws2}$ (MW) | 30 | 30 | 26.0320 | 17.1734 | 30 | 22.964 |
| $Q_{ss}$ (MW) | 20.0658 | 17.2351 | 20.1288 | 17.6723 | 25 | 8.0319 |
| Cost (\$/h) | **804.2061** | 804.3160 | 804.4383 | 805.7037 | 806.0331 | 804.8684 |
| Emission (t/h) | **0.4994** | 0.5011 | 0.5016 | 0.5273 | 0.5362 | 0.5151 |

4.3.2. Case 2: The System Only Incorporating Stochastic Wind

In this case study, the test system only contains two wind farms and does not include solar PV. For the sake of detailed comparison with two methods presented in literature [61], three different solutions of MMOCE are selected, and the simulation results are listed in Table 17. It is clear that the proposed MMOCE can find the lowest values of cost and emission, which means it can find better solutions than CMOPEO-EED and CNSGAII-EED. Figure 12 compares the Pareto fronts between MMOCE and competitor algorithms, and the compromise solutions are listed in Table 18. The data curves show that the proposed MMOCE converges closer to optimal solutions than other algorithms. Furthermore, the result obtained by MMOCE, with 834.5583 $/h in cost and 0.5896 t/h in emission, stands out from all algorithms.

**Table 17.** Simulation results obtained by MMOCE, CMOPEO-EED, and CNSGAII-EED in Case 2.

| Control Variables | Cost-Solution (A0, A1, A2) | | | Emission-Solution (B0, B1, B2) | | | Compromise-Solution (C0, C1, C2) | | |
|---|---|---|---|---|---|---|---|---|---|
| | MMOCE | CMOEO-EED [61] | CNSGAII-EED [62] | MMOCE | CMOEO-EED [61] | CNSGAII-EED [62] | MMOCE | CMOEO-EED [61] | CNSGAII-EED [62] |
| $P_{TG_1}$ (MW) | 133.0246 | 133.4204 | 136.1377 | 70.7194 | 66.0442 | 71.0043 | 114.3254 | 114.3152 | 117.6553 |
| $P_{TG_2}$ (MW) | 41.6542 | 36.4690 | 32.7275 | 58.6058 | 60.0168 | 57.9792 | 55.9144 | 49.4717 | 48.4645 |
| $P_{TG_3}$ (MW) | 10 | 11.0759 | 20.4909 | 24.2350 | 35 | 31.2133 | 10.3215 | 11.3607 | 22.6222 |
| $P_{TG_4}$ (MW) | 12 | 12 | 13.1202 | 23.6914 | 12 | 21.8278 | 12 | 12 | 13.1951 |
| $P_{ws1}$ (MW) | 49.5673 | 52.6521 | 50.1389 | 60.8875 | 62.1738 | 57.9794 | 52.9072 | 57.7254 | 50.1389 |
| $P_{ws2}$ (MW) | 42.9147 | 43.4150 | 36.7650 | 48.4544 | 51.1847 | 46.9542 | 43.0633 | 43.4150 | 36.7531 |
| $V_1$ (p.u.) | 1.0720 | 1.0638 | 1.0903 | 1.0652 | 1.0626 | 1.0817 | 1.0694 | 1.0667 | 1.0903 |
| $V_2$ (p.u.) | 1.0591 | 1.0530 | 0.9701 | 1.0587 | 1.0530 | 0.9705 | 1.0558 | 1.0584 | 0.9712 |
| $V_5$ (p.u.) | 1.0388 | 1.1000 | 1.0944 | 1.0396 | 1.1000 | 1.1000 | 1.0379 | 1.1000 | 1.0837 |
| $V_8$ (p.u.) | 1.0553 | 1.0650 | 0.9875 | 1.0565 | 1.0497 | 0.9905 | 1.0574 | 1.0585 | 0.9855 |
| $V_{11}$ (p.u.) | 1.0800 | 1.0359 | 1.0726 | 1.0926 | 1.0424 | 1.0737 | 1.1000 | 1.0441 | 1.0727 |
| $V_{13}$ (p.u.) | 1.0511 | 1.1000 | 1.0555 | 1.0510 | 1.1000 | 1.0574 | 1.0510 | 1.1000 | 1.0567 |
| $Q_{TG_1}$ (MW) | −4.0466 | −11.6123 | 45.3146 | −4.2832 | 2.9649 | 38.4253 | 1.3983 | −11.3175 | 44.8153 |
| $Q_{TG_2}$ (MW) | 15.6637 | 13.3567 | −20 | 11.4685 | −3.9619 | −20 | 3.8894 | 12.5536 | −20 |
| $Q_{TG_3}$ (MW) | 40 | 40 | 40 | 40 | 40 | 40 | 40 | 40 | 40 |
| $Q_{TG_4}$ (MW) | 18.2106 | 23.0313 | −1.6361 | 16.0053 | 16.6208 | −1.9824 | 17.5542 | 19.1638 | −2.8490 |
| $Q_{ws1}$ (MW) | 24.1201 | 23.7141 | 35 | 18.9026 | 25.1861 | 35 | 23.8134 | 25.3184 | 35 |
| $Q_{ws2}$ (MW) | 24.2296 | 30 | 20.0770 | 27.9365 | 30 | 20.2949 | 30 | 29.8266 | 20.0146 |
| Cost ($/h) | **810.3276** | 810.6380 | 814.6888 | **888.5777** | 888.5999 | 888.9853 | **834.5583** | 834.6694 | 835.5906 |
| Emission | **1.6216** | 1.6618 | 1.9532 | **0.1645** | 0.1664 | 0.1706 | **0.5896** | 0.5899 | 0.6924 |

**Table 18.** Comparison of the compromise solutions obtained by all algorithms in Case 2.

| | MMOCE | DMOCE | FMOCE | MOO CEM | NSGAII | MOPSO |
|---|---|---|---|---|---|---|
| $P_{TG_1}$ (MW) | 114.3254 | 114.6435 | 114.4487 | 114.6034 | 117.1215 | 114.9084 |
| $P_{TG_2}$ (MW) | 55.9144 | 57.6041 | 50.0948 | 50.9100 | 42.1368 | 53.3999 |
| $P_{TG_3}$ (MW) | 10.3215 | 10 | 13.4001 | 15.1987 | 17.4805 | 10.0105 |
| $P_{TG_4}$ (MW) | 12 | 12 | 12 | 12 | 16.3096 | 12.7678 |
| $P_{ws1}$ (MW) | 52.9072 | 51.6911 | 55.0908 | 50.1419 | 51.6231 | 52.8663 |
| $P_{ws2}$ (MW) | 43.0633 | 42.7644 | 43.4177 | 46.0267 | 43.8745 | 44.7722 |
| $V_1$ (p.u.) | 1.0694 | 1.0718 | 1.0737 | 1.0511 | 1.0543 | 1.1000 |
| $V_2$ (p.u.) | 1.0558 | 1.0138 | 0.9647 | 1.0385 | 1.0298 | 1.0741 |
| $V_5$ (p.u.) | 1.0379 | 1.0457 | 1.0556 | 1.0337 | 1.0506 | 1.0556 |
| $V_8$ (p.u.) | 1.0574 | 1.1000 | 1.0828 | 1.0500 | 1.0765 | 1.0899 |
| $V_{11}$ (p.u.) | 1.1000 | 1.0917 | 1.0958 | 0.9836 | 1.0723 | 1.0126 |
| $V_{13}$ (p.u.) | 1.0510 | 1.0639 | 1.0304 | 1.0408 | 1.0637 | 1.0419 |
| $Q_{TG_1}$ (MW) | 1.3983 | 13.6363 | 20.6437 | 1.8182 | 10.1473 | 36.0754 |
| $Q_{TG_2}$ (MW) | 3.8894 | −20 | −20 | 14.1036 | −20 | −6.2083 |
| $Q_{TG_3}$ (MW) | 40 | 40 | 40 | 40 | 40 | 40 |
| $Q_{TG_4}$ (MW) | 17.5542 | 22.2206 | 10.8751 | 37.3132 | 27.5411 | 15.4431 |
| $Q_{ws1}$ (MW) | 23.8134 | 33.8238 | 35 | 35 | 35 | 29.3385 |
| $Q_{ws2}$ (MW) | 30 | 27.2668 | 30 | 1.7752 | 24.8045 | 1.4720 |
| Cost ($/h) | **834.5583** | 834.7172 | 834.9402 | 836.5329 | 834.8781 | 834.6457 |
| Emission (t/h) | **0.5896** | 0.5990 | 0.5919 | 0.5960 | 0.6710 | 0.6056 |

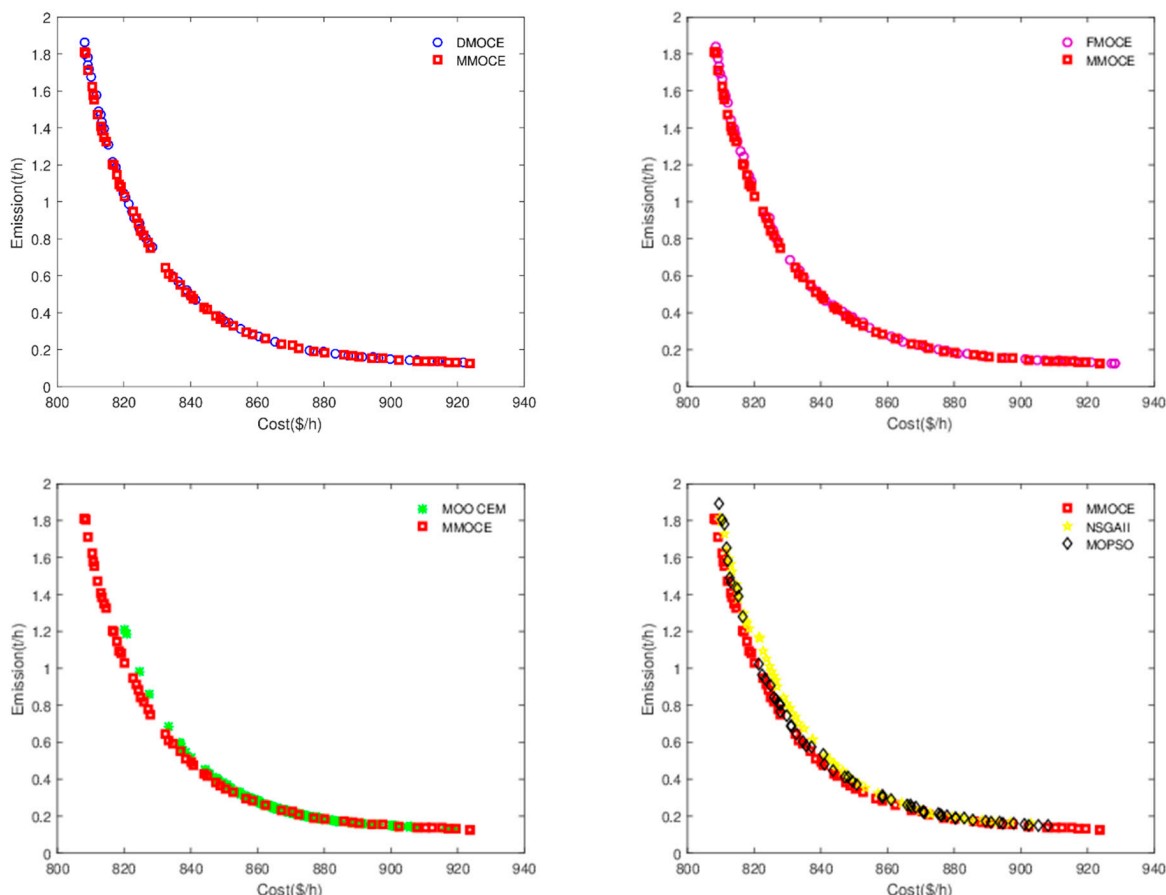

**Figure 12.** Pareto front of MMOCE and other algorithms in Case 2.

4.3.3. Case 3: The System Only Incorporating Stochastic Solar PV

The test system in this case contains only solar PV. Similar to the above two cases, the compromise solution of the proposed MMOCE is selected to compared with the algorithms in [63], and Table 19 lists the results clearly. Figure 13 depicts the Pareto front of MMOCE and other algorithms in Case 3. It is not hard to see that MMOCE still has the merits in convergence and effectiveness.

**Table 19.** Simulation results obtained by MMOCE, CMOPEO-EED and CNSGAII-EED in Case 3.

| Control Variables | Cost-Solution (A0, A1, A2) | | | Emission-Solution (B0, B1, B2) | | | Compromise-Solution (C0, C1, C2) | | |
|---|---|---|---|---|---|---|---|---|---|
| | MMOCE | CMOEO-EED [61] | CNSGAII-EED [62] | MMOCE | CMOEO-EED [61] | CNSGAII-EED [62] | MMOCE | CMOEO-EED [61] | CNSGAII-EED [62] |
| $P_{TG_1}$ (MW) | 134.0754 | 134.0858 | 139.4682 | 87.1357 | 88.4465 | 89.6932 | 117.5678 | 117.9506 | 119.7446 |
| $P_{TG_2}$ (MW) | 61.5611 | 60.6874 | 57.1434 | 66.5074 | 77.4796 | 75.9446 | 63.4474 | 70.2428 | 67.2314 |
| $P_{TG_3}$ (MW) | 20.6333 | 13.9832 | 10.1346 | 22.7809 | 35 | 26.7500 | 21.0568 | 21.4771 | 19.5729 |
| $P_{TG_4}$ (MW) | 15 | 21.5477 | 20.2378 | 50 | 25.1678 | 29.9591 | 23.1620 | 21.5477 | 19.4187 |
| $P_{TG_5}$ (MW) | 10 | 11.5842 | 17.4797 | 12.4370 | 12.7764 | 19.9521 | 14.6247 | 10 | 17.4850 |
| $P_{ss}$ (MW) | 49.7022 | 49.0552 | 47.1022 | 49.7600 | 50 | 46.8430 | 50 | 49.0552 | 47.2479 |
| $V_1$ (p.u.) | 1.0686 | 1.0658 | 1.0654 | 1.0607 | 1.0629 | 1.0654 | 0.9789 | 1.0658 | 1.0654 |
| $V_2$ (p.u.) | 1.0578 | 1.0537 | 0.9932 | 1.0530 | 1.0532 | 0.9945 | 1.1000 | 1.0537 | 0.9944 |
| $V_5$ (p.u.) | 1.0307 | 1.1000 | 1.0410 | 1.0280 | 1.1000 | 1.0291 | 1.0379 | 1.1000 | 1.0409 |
| $V_8$ (p.u.) | 1.0403 | 1.0258 | 1.0413 | 1.0408 | 1.0317 | 1.0519 | 1.1000 | 1.0306 | 1.0513 |
| $V_{11}$ (p.u.) | 1.1000 | 1.1000 | 1.0324 | 1.0994 | 1.1000 | 1.0326 | 1.0713 | 1.1000 | 1.0324 |
| $V_{13}$ (p.u.) | 1.0738 | 1.0752 | 1.0946 | 1.0773 | 1.1000 | 1.0914 | 1.1000 | 1.1000 | 1.0977 |
| $Q_{TG_1}$ (MW) | −11.3892 | −9.2501 | 14.6380 | −7.5218 | −4.5573 | 15.9405 | 1.6784 | −5.8103 | 8.3678 |
| $Q_{TG_2}$ (MW) | 16.9285 | 12.4732 | −20 | 11.6874 | −0.4635 | −20 | −20 | 3.7074 | −20 |
| $Q_{TG_3}$ (MW) | 26.7988 | 40 | 40 | 26.4980 | 40 | 28.4233 | 40 | 40 | 40 |
| $Q_{TG_4}$ (MW) | 36.5410 | 24.5279 | 53.3713 | 29.9588 | 27.1887 | 56.3327 | 62.5000 | 28.2403 | 58.4254 |
| $Q_{TG_5}$ (MW) | 29.0576 | 29.5347 | 10.7791 | 29.0230 | 28.0240 | 9.7916 | 6.8007 | 28.7868 | 8.7660 |
| $Q_{ss}$ (MW) | 24.1922 | 25 | 25 | 25 | 30 | 25 | 23.5308 | 25 | 25 |
| Cost ($/h) | **811.7028** | 812.1683 | 813.8361 | **876.5106** | 877.6596 | 883.3316 | **836.1290** | 836.2670 | 836.4149 |
| Emission | **1.7444** | 1.7453 | 2.4089 | **0.2429** | 0.2539 | 0.2576 | **0.7172** | 0.7246 | 0.7897 |

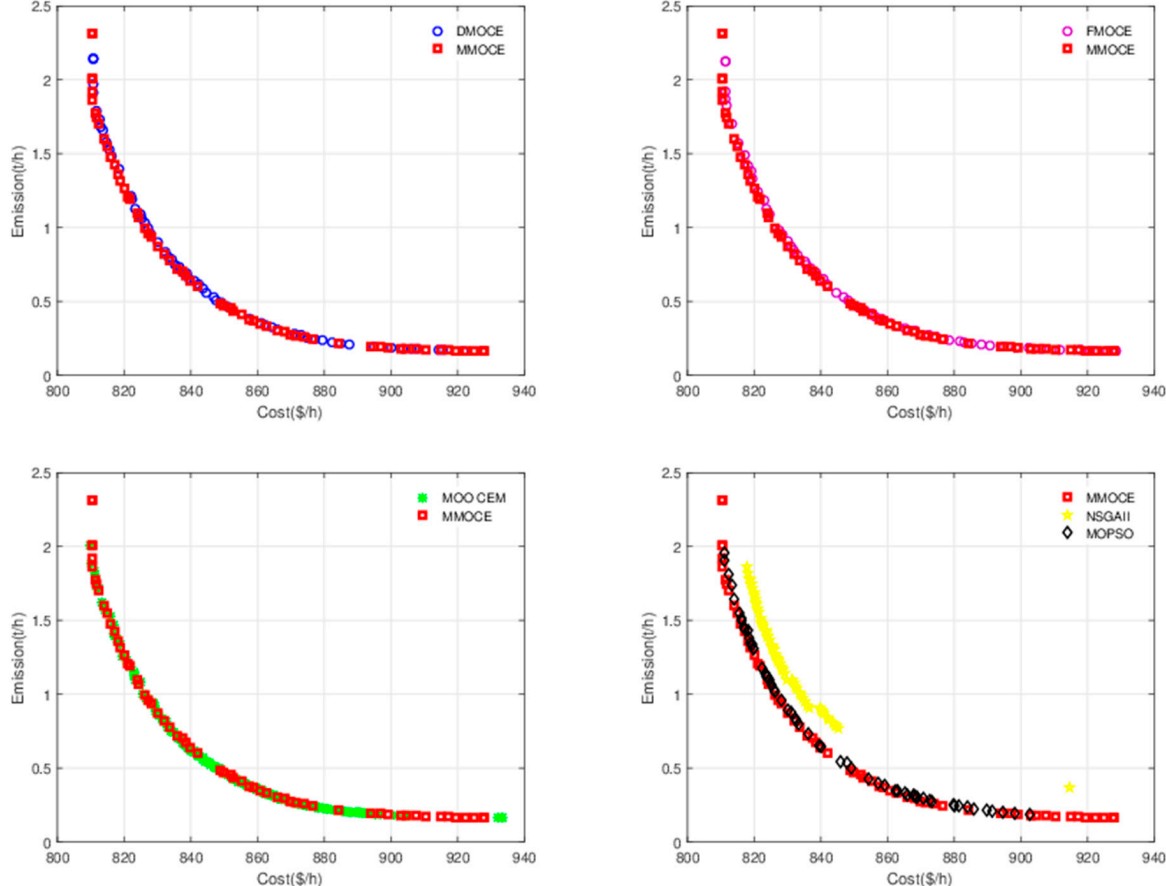

**Figure 13.** Pareto front of MMOCE and other algorithms in Case 3.

Furthermore, the comparison between MMOCE and other competitor algorithms is conducted to verify the superiority of MMOCE while the results are listed in Table 20. Though it does not demonstrate evident advantage over MOO CEM with 836.1867 $/h in

cost and 0.7107 t/h in emission in terms of performance, a satisfactory result obtained by MMOCE is better than other algorithms.

**Table 20.** Comparison of the compromise solutions obtained by all algorithms in Case 3.

|  | MMOCE | DMOCE | FMOCE | MOO CEM | NSGAII | MOPSO |
|---|---|---|---|---|---|---|
| $P_{TG_1}$ (MW) | 117.5678 | 118.1924 | 117.9340 | 117.6791 | 122.5261 | 118.1972 |
| $P_{TG_2}$ (MW) | 63.4474 | 64.6039 | 71.7869 | 65.0059 | 51.5835 | 68.4740 |
| $P_{TG_3}$ (MW) | 21.0568 | 17.8186 | 21.4148 | 21.6037 | 23.0837 | 18.7274 |
| $P_{TG_4}$ (MW) | 23.1620 | 27.5397 | 19.8652 | 20.1938 | 46.0255 | 23.7451 |
| $P_{ws1}$ (MW) | 14.6247 | 12.4316 | 10 | 15.4506 | 11.9970 | 11.2675 |
| $P_{ws2}$ (MW) | 50 | 49.7286 | 49.5265 | 50 | 34.5972 | 50 |
| $V_1$ (p.u.) | 0.9789 | 1.0679 | 1.0648 | 1.1000 | 1.0687 | 1.0686 |
| $V_2$ (p.u.) | 1.1000 | 0.9500 | 1.0519 | 1.0164 | 1.0309 | 0.9635 |
| $V_5$ (p.u.) | 1.0379 | 1.1000 | 1.0159 | 1.0798 | 1.0308 | 1.0971 |
| $V_8$ (p.u.) | 1.1000 | 1.0387 | 1.0927 | 1.1000 | 1.0409 | 1.0446 |
| $V_{11}$ (p.u.) | 1.0713 | 1.1000 | 1.0593 | 1.0663 | 1.0999 | 1.0998 |
| $V_{13}$ (p.u.) | 1.1000 | 1.0995 | 1.0453 | 1.0714 | 1.0137 | 1.0654 |
| $Q_{TG_1}$ (MW) | 1.6784 | 11.5304 | −2.6881 | 9.5990 | 20.7913 | 8.1282 |
| $Q_{TG_2}$ (MW) | −20 | −20 | 10.6724 | −20 | −20 | −20 |
| $Q_{TG_3}$ (MW) | 40 | 40 | 15.8632 | 40 | 35.9355 | 40 |
| $Q_{TG_4}$ (MW) | 62.5000 | 34.1579 | 62.5000 | 62.5000 | 44.5888 | 42.1869 |
| $Q_{ws1}$ (MW) | 6.8007 | 29.6849 | 17.1431 | 8.3374 | 33.3207 | 29.0461 |
| $Q_{ws2}$ (MW) | 23.5308 | 25 | 16.7957 | 14.8923 | 4.6421 | 21.0298 |
| Cost ($/h) | **836.1290** | 836.7103 | 836.9941 | 836.1867 | 836.4888 | 836.3250 |
| Emission (t/h) | **0.7172** | 0.7303 | 0.7241 | 0.7107 | 0.9077 | 0.7322 |

### 4.4. Combined Emission Economic Dispatch Problems with Wind Penetration

In this paper, the test system in this section consists of 40 thermal units and a wind conversion device. The characteristic data of the 40-unit system are given in [64]. The B matrix coefficients are taken from [63], since the transmission loss is taken into consideration. The wind power penetration is set to 10%, and $P_d$ is 10,500MW. As the system grows in size, the optimization becomes more difficult. In this paper, several multi-objective algorithms (i.e., MMOCE, DMOCE, FMOCE, NSGAII, MOPSO, MOO CEM) are employed to optimize this system. Figure 14 presents the optimal Pareto solutions obtained by these multi-objective algorithms, and the compromise solution obtained by MMOCE is listed in Table 21. Table 22 shows the comparison performance between the compromise solutions obtained by these algorithms.

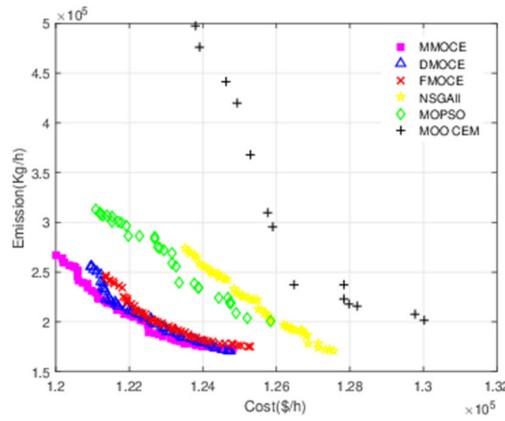

**Figure 14.** The Pareto front of various algorithms for 40-unit system.

**Table 21.** The compromise solution obtained by MMOCE.

| Unit | Output (MW) | Unit | Output (MW) | Unit | Output (MW) | Unit | Output (MW) |
|---|---|---|---|---|---|---|---|
| 1 | 113.8225 | 11 | 164.8471 | 21 | 438.6464 | 31 | 179.5618 |
| 2 | 114 | 12 | 94.6529 | 22 | 495.7881 | 32 | 185.4391 |
| 3 | 120 | 13 | 398.4712 | 23 | 441.9609 | 33 | 187.7371 |
| 4 | 179.4560 | 14 | 401.3060 | 24 | 434.6433 | 34 | 199.9304 |
| 5 | 97 | 15 | 390.9459 | 25 | 495.3594 | 35 | 193.1095 |
| 6 | 105.9222 | 16 | 394.2235 | 26 | 437.4695 | 36 | 200 |
| 7 | 300 | 17 | 488.5986 | 27 | 12.5754 | 37 | 98.2511 |
| 8 | 289.0161 | 18 | 401.9877 | 28 | 10.5770 | 38 | 110 |
| 9 | 286.9929 | 19 | 423.0220 | 29 | 12.5574 | 39 | 109.9946 |
| 10 | 205.0808 | 20 | 424.7560 | 30 | 87.5222 | 40 | 503.9188 |

**Table 22.** Comparison of the compromise solutions obtained by all algorithms in the third model.

| Algorithms | Fuel Cost ($/h) | Emission (Kg/h) |
|---|---|---|
| MMOCE | 122,018.6133 | 207,075.3120 |
| FMOCE | 122,292.4613 | 207,228.6658 |
| DMOCE | 122,453.2896 | 207,206.8501 |
| MOO CEM | 126,478.3572 | 237,218.5618 |
| NSGAII | 125,509.5808 | 212,384.3552 |
| MOPSO | 123,163.8923 | 258,994.9946 |

From Figure 14, it is shown that the proposed MMOCE is superior to all other competitor algorithms in terms of the convergence. Table 22 lists their compromise solutions, and it is clearly shown that the solution obtained by MMOCE represents a cost of 122,018.6133 $/h and emission of 207,075.3012 Kg/h, which is better than other solutions obtained by other algorithms. It implies that when it is applied to large scale optimization systems, MMOCE is still able to exhibit good convergence and has the potential to provide a competitive solution for the decision maker. Compared with other multi-objective cross entropy methods, the proposed MMOCE dramatically improves the global search ability for large scale systems and exhibits excellent competitiveness.

To examine the convergence performance of the proposed method, the convergence trends for the minimum cost and minimum emissions of MMOCE and other algorithms are shown in Figure 15. The max function evaluation is set to 20,000. According to Figure 15a, the minimum cost of MMOCE decreases quickly in the first 410 iterations, and it tends to be stable after 800. In Figure 15b, the minimum emission of MMOCE decreases quickly in the first 180 iterations. The emission tends to be stable after 200 iterations. Based on Figure 15a,b, MMOCE converges faster than other methods.

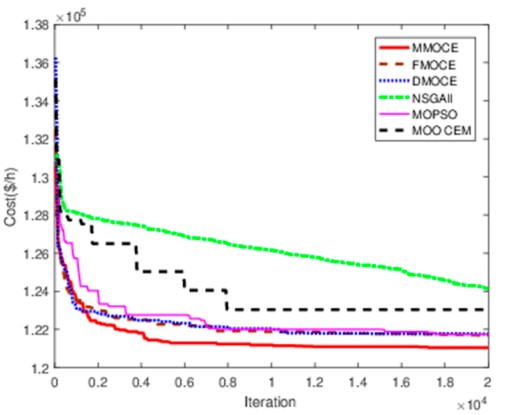

(**a**) Convergence property for minimum cost

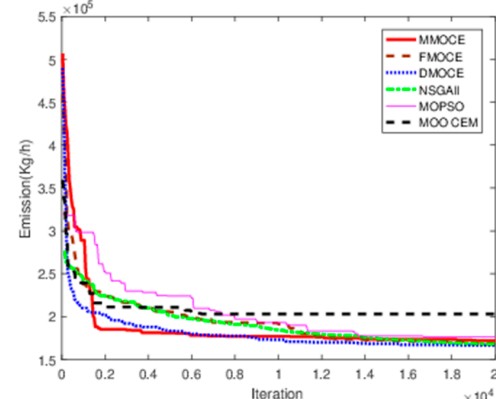

(**b**) Convergence property for minimum emission

**Figure 15.** Convergence property for minimum cost and minimum emission of all methods.

## 5. Conclusions

In this paper, a modified multi-objective cross entropy algorithm is proposed based on the conventional cross entropy method. The drawbacks of the conventional CE method have been effectively tackled, and the optimal performance is improved to a great extent. To evaluate its performance, the proposed method is tested on eight well-known benchmark functions. The numerical results are compared with the different heuristic optimization algorithms, which confirms the performance of the proposed method in terms of the convergence and diversity. In addition, the proposed algorithm is used to optimize three different EED problems with renewable energy sources.

The first two models (i.e., CHEED system and the modified IEEE 30-bus and six-generator system) consider the availability of wind and solar power, and the comparisons with other optimization methods reveal that the proposed method significantly outperformed other methods in all cases, confirming that the proposed method is capable of enhancing the convergence and seeking the optimal global solution. The third 40-unit system uses wind power as negative loads, and the numerical results also exhibit better convergence characteristics for large-scale test systems. In summary, the test results confirm the feasibility and superiority of the proposed MMOCE method.

The future work will include further simplification of the parameters used in the conventional CE method and improvement of the scalability of the method for optimizing multi-objective engineering optimization problems. Additionally, in this paper, three different types of EED model are tested. However, the renewable sources of the first model and the third model are deterministic and are only regarded as negative loads. This handling approach for renewable sources is simple and neglects the stochastic nature of the renewable resources. The future work will explore EED model considering the stochastic characteristics of various intermittent energy sources.

**Author Contributions:** Methodology: Q.N.; supervision: Q.N. and Z.Y.; writing—original draft: M.Y. and Y.Z.; writing—review and editing: M.Y. All authors have read and agreed to the published version of the manuscript.

**Funding:** This research is financially supported by China NSFC (61773252, 62003332) and Outstanding Young Researcher Innovation Fund of Shenzhen Institute of Advanced Technology, Chinese Academy of Sciences (201822).

**Institutional Review Board Statement:** Not applicable.

**Informed Consent Statement:** Not applicable.

**Data Availability Statement:** No new data were created or analyzed in this study. Data sharing is not applicable to this article.

**Acknowledgments:** There is no any support which is not covered by the author contribution or funding sections.

**Conflicts of Interest:** The authors declare no conflict of interest.

## Nomenclature

The following abbreviations are used in this manuscript:

| | |
|---|---|
| $C_{total}$ | Total production cost |
| $E_{total}$ | Total emission |
| $C_i(P_i)$ | Production cost of thermal power unit $i$ |
| $C_j\left(P_j, H_j\right)$ | Production cost of cogeneration unit $j$ |
| $C_k(H_k)$ | Production cost of heat-only unit $k$ |
| $E_{pi}(P_i)$ | Emission of thermal power unit $i$ |
| $E_{cj}\left(P_j, H_j\right)$ | Emission of cogeneration unit $j$ |
| $E_{hk}(H_k)$ | Emission of heat-only unit $k$ |

| | |
|---|---|
| $P_d, P_L$ | Load demand and transmission loss |
| $P_{wind}, P_{solar}$ | Wind power and solar power |
| $H_D$ | Total heat demand |
| $P_i^{max}, P_i^{min}$ | Power capacity limits of thermal power unit $i$ |
| $H_k^{max}, H_k^{min}$ | Heat capacity limits of heat-only unit $k$ |
| $N_p$ | Number of thermal power units |
| $N_c$ | Number of cogeneration units |
| $N_h$ | Number of heat-only units |
| $a_i, b_i, d_i$ | Cost coefficients for thermal power unit $i$ |
| $m_i, n_i, l_i, x_i, y_i, z_i$ | Cost coefficients for cogeneration unit $j$ |
| $\alpha_k, \beta_k, \gamma_k$ | Cost coefficients for heat-only unit $k$ |
| $\delta_i, \epsilon_i, \xi_i$ | Emission coefficients for thermal power unit $i$ |
| $\mu_j$ | Emission coefficients for cogeneration unit $j$ |
| $\sigma_k$ | Emission coefficients for heat-only unit $k$ |
| $N_{TU}$ | Total number of thermal power units |
| $P_{TU}$ | Power produced by generating unit $j$ |
| $\alpha_j, \beta_j, \gamma_j$ | Cost coefficients for thermal power unit $j$ |
| $d_j, e_j$ | Valve effect coefficients for thermal power unit $j$ |
| $P_{TU}^{min}$ | Minimum power of thermal power unit $j$ |
| $h_i$ | Cost constant of wind power unit $i$ |
| $P_{sw,i}$ | Scheduled wind power from unit $i$ |
| $K_{RC,i}$ | Reverse cost coefficient of wind unit $i$ |
| $P_{wa,i}$ | Actual available power of wind unit $i$ |
| $f_{wp}(P_{w,i})$ | Wind power probability function of wind unit $i$ |
| $K_{PC,i}$ | Penalty cost coefficient of wind unit $i$ |
| $P_{rw,i}$ | Rated output power of wind unit $i$ |
| $V_{out}, V_r, V_{in}$ | Cut out wind speed, nominal wind speed, and cut in wind speed |
| $P_{rw}$ | Rated power |
| $v$ | Current wind speed |
| $g_i$ | Cost constant of solar power unit $j$ |
| $P_{ss,j}$ | Scheduled solar power from unit $j$ |
| $K_{SR,j}$ | Reverse cost coefficient of solar power unit $j$ |
| $P_{sa,j}$ | Actual available power of solar power unit $j$ |
| $K_{PS,j}$ | Penalty cost coefficient of solar power unit $j$ |
| $p_i, q_i, r_i, s_i, t_i$ | Emission coefficients for thermal power unit $i$ |

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
