# Peer review of "Economic Emission Dispatch Considering Renewable Energy Resources—A Multi-Objective Cross Entropy Optimization Approach"

_sustainability, doi:10.3390/su13105386_

Round 1

Reviewer 1 Report

Dear Authors,

I have carefully and critically read the manuscript.

(a) Authors need to list the list of abbreviations of all short forms they used in their manuscript in the Nomenclature for greater readability.

(b) Conclusion's second paragraph sentence is vague, I ask authors to rephrase and improve its meaning for readers.

Reviewer 2 Report

This is a well-written paper and this reviewer enjoyed reading this work. I only have some comments:

1) in equation 23, there is something missing in the denominator; 

2) The authors should discuss more in detail the drawbacks and limitations of their model. 

3) Why is the population size chosen as 100 (L365)?

4) It would be nice to mention the overall simulation time. 

5) What is the definition of emission in your work? Is it only coming from NOx and SOx? How about if one wanted to incorporate the emissions from carbon-based species?
